# A view of the European carbon flux landscape through the lens of the ICOS atmospheric observation network

Ida Storm[1], Ute Karstens[1], Claudio D'Onofrio[1], Alex Vermeulen[1,2], Wouter Peters[3,4]

[1]ICOS Carbon Portal at Lund University, Department of Physical Geography and Ecosystem Sciences, Lund, Sweden
[2]ICOS ERIC, Carbon Portal, Lund, Sweden
[3]Wageningen University, Environmental Sciences Group, Wageningen, The Netherlands
[4]University of Groningen, Centre for Isotope Research, Groningen, The Netherlands

*Correspondence to*: Ida Storm (ida.storm@nateko.lu.se)

**Abstract.** The ICOS (Integrated Carbon Observation System) network of atmospheric measurement stations produces standardized data on greenhouse gas concentrations at 46 stations in 16 different European countries (March 2023). The placement of instruments on tall towers and mountains makes for large influence regions ("concentration footprints"). The combined footprints for all the individual stations create a "lens" through which the network sees the European $CO_2$ flux landscape. In this study, we summarize this view using quantitative metrics of the fluxes seen by individual stations, and by the current and extended ICOS network. Results are presented both from a country-level and pan-European perspective, using open-source tools that we make available through the ICOS Carbon Portal. We target anthropogenic emissions from various sectors, as well as the land cover types found across Europe and their spatiotemporally varying fluxes. This recognizes different interests of different ICOS stakeholders. We specifically introduce "monitoring potential maps" to identify which regions have a relative underrepresentation of biospheric fluxes. This potential changes with the introduction of new stations, which we investigate for the planned ICOS expansion with 19 stations over the next few years.

In our study focused on the summer of 2020, we find that the ICOS atmospheric station network has limited sensitivity to anthropogenic fluxes, as was intended in the current design. Its representation of biospheric fluxes follows the fractional representation of land cover and is generally well-balanced considering the pan-European view. Exceptions include representation of grass & shrubland and broadleaf forest which are abundant in south-eastern European countries, particularly Croatia and Serbia. On country scale the representation shows larger imbalances, even within relatively densely monitored countries. The flexibility to consider both individual ecosystems, countries, or their integrals across Europe demonstrates the usefulness of our analyses and can readily be re-produced for any network configuration within Europe.

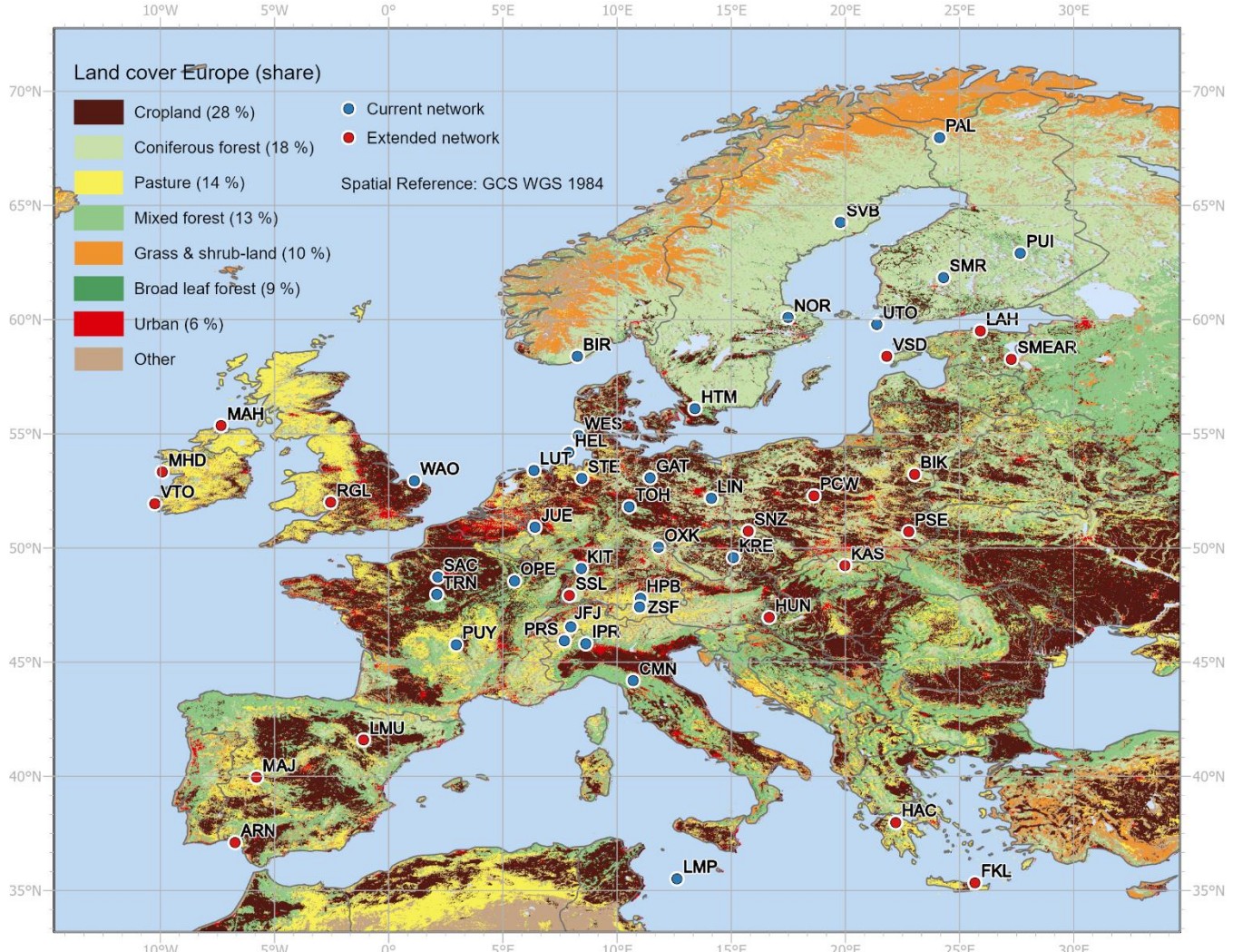

**Figure 1: Current (November 2022) and prospective ICOS atmospheric stations within the STILT model domain (see Sect. 2.1). Main land cover types (HILDA; Winkler et al., 2020) and their total shares given countries contained within the model domain are shown in the legend.**

# 1 Introduction

Rising levels of carbon dioxide ($CO_2$) and its forcing towards a warmer climate have led 195 countries to sign the Paris Agreement, which was adopted in 2015. Countries committed to reduce their emissions and to review their commitments every five years in response to a common $CO_2$ trajectory. Up until now, about half of the $CO_2$ humans have emitted has been taken up by land (29% of total $CO_2$ emissions 2011-2020, Friedlingstein et al., 2022) or stored in the deep ocean (26% of total $CO_2$ emissions 2011-2020, Friedlingstein et al., 2022). The other half of the anthropogenic $CO_2$ remains in the atmosphere and contributes to the atmospheric growth rate which is 2.5 ppm for 2022 according to a preliminary estimate by Friedlingstein et

al. (2022). On a global scale, the common $CO_2$ trajectory will greatly depend on the capacity for storage in these carbon

reservoirs and it will be important to plan our efforts under the Paris agreement. Furthermore, understanding the natural carbon exchanges between carbon reservoirs is important for our ability to track and verify changes in emissions (Balsamo et al., 2021).

Our understanding of the carbon cycle has evolved over the last few decades, and atmospheric observations have been

indispensable to gain deeper insights (Tans et al., 1990; Keeling et al., 2001; Francey et al., 1999; Bacastow et al., 1985). Long-standing records of direct $CO_2$ measurements, e.g. the canonical one from Mauna Loa, Hawaii, show the increasing trend in global $CO_2$ levels (Sundquist and Keeling, 2009), and continue to form the basis of long-term analyses (Ballantyne et al., 2012; Graven et al., 2013; Liu et al., 2020). Additionally, inverse modelling systems have been employed at various scales to balance the atmospheric carbon budget, ensuring its consistency with observations from worldwide monitoring networks

(Peylin et al., 2013; Thompson et al., 2016; Gaubert et al., 2019). Because of the relatively high uncertainty of biosphere fluxes compared to anthropogenic emissions, such studies have generally focused on exchange of $CO_2$ with the biosphere and oceans.

Measurements of atmospheric mole fractions have traditionally been collected at remote islands, mountain tops, or other locations at large distance from direct emissions or uptake, to find well-mixed conditions that represent background

atmospheric levels (Conway et al., 1994). Regional networks were added to this in the last two decades, with tall towers and aircraft data to inform on continental gradients in emissions and uptake (Sweeney et al., 2015; Turnbull et al., 2018). The European monitoring capacity is currently organized through the Integrated Carbon Observation System (ICOS; Heiskanen et al., 2021), a European infrastructure that provides standardized data on greenhouse gas concentrations in the atmosphere and fluxes between the atmosphere, land, and oceans. The ICOS network of atmospheric stations currently includes 46 stations in

14 countries (status of March 2023).

The measurements from the ICOS atmospheric stations target a strongly heterogeneous flux landscape; Europe has multiple climate zones ranging from Mediterranean in the South with, according to the Köppen-Geiger classification, dry and hot summers, through temperate, to cold Northern sub-Arctic climate without a dry season (Beck et al., 2018). The main land

cover types are cropland, coniferous forest, pasture, mixed forest, grass & shrubland and broadleaf forest (see Fig. 1). Coniferous forest and grass & shrubland are prominent in the north, whereas the rest of Europe is more heterogeneous and generally dominated by cropland. Ecosystem management is strong across Europe, with land-use history, forest management, and cultivation of grasslands, croplands, and wetlands showing large differences from country to country. As a result of this heterogeneity, different ecosystems have different responses to climate anomalies, such as drought.


The network's ability to inform on the carbon cycle, such as responses to the 2018 drought (Peters et al., 2020; Ramonet et al., 2020), is directly tied to the influence areas ("concentration footprints") of its stations. A station footprint represents where the

air has passed on the way to the station for a specific point in time, and the carbon exchange in the footprint area is expected to influence the concentration at the station. Analyses of footprints to understand station and network representations have been exploited in previous studies. For example, Oney et al. (2015) used station footprints to analyse the suitability of a Swiss network of four stations for regional-scale carbon flux studies. A visual inspection of the average station footprints, and the expected signals associated with different land cover types, supported claims about where the monitoring can be expected to provide useful information. In Henne et al. (2010), footprints were analysed to classify stations based on expected representativeness of their measurements. Representative measurements have little or no influence from local emission sources, which make them appropriate for inclusion in regional inversion studies. For ecosystem sites, where fluxes rather than concentrations are measured, the "flux footprints" are small with influence mainly from the site's immediate surrounding. In this context, the idea of representation has been applied in Pallandt et al. (2022) to assess what Arctic ecosystem types are at the site locations compared to what ecosystems are found in the Arctic. Malone et al. (2022) similarly identified gaps in the U.S. NEON network based on representation of different clusters identified based on their ecological properties. In both studies, the evaluation of the network representation was subsequently used to advise on future expansion and the appropriateness of upscaling of the fluxes to larger regions.

Previous network design studies also employed Quantitative Network Design (QND), where the impact of a given set of existing or hypothetical observations in a modelling framework is assessed to find an optimal network for a selected study area (Kaminski and Rayner, 2017). The metric of how much value a potential station adds is typically the reduction in the assumed a-priori uncertainties of the carbon fluxes. QND often results in small networks targeting the largest signals, or on specific sources assumed least well-known. For example, in a QND study by Nickless et al. (2020) for Africa they found the optimal network to be focused on the productive region around the equator and that it changes with the seasons reflecting difference in flux activity and hence uncertainty. The chosen model set-up, the freedom to choose new station locations, and how the uncertainties are prescribed to the flux landscape thus has an influence on the results, and there is no fully objective quantification of optimal network design.

In this study, we combine footprint analyses and quantification of sensing capacity using a different approach. Similar to the mentioned ecosystem studies, it focuses on what is seen by a station or network relative to the regional or national flux landscape, or relative to other stations, networks, or countries in Europe. Any underlying spatial data layer, ranging from population density to forest age or anthropogenic carbon emission, can be part of this quantification, recognizing that some fluxes, such as forests with high potential for long-term storage of carbon, might be more important than others. Our approach hence allows for flexibility in defining what makes for an appropriate station location, or an appropriate network, and allows expert judgement or formal optimization based on the outcomes.

We chose in this study to quantify and summarize the capacity of the ICOS atmospheric observing network to sense the underlying $CO_2$ flux landscape. Other mole fractions observed by the network including $CH_4$ and $N_2O$ exhibit different flux distributions and would therefore need separate analyses to characterise their gaps and monitoring potentials. We include anthropogenic emissions as well as biospheric fluxes, and we discuss scales from individual stations, to countries, and to pan-European fluxes. Our main research goal is to identify areas with unexploited "monitoring potential", a novel concept that we introduce in this work. Areas with relatively high monitoring potential with respect to a specific ecosystem type would likely return useful information if targeted by an expansion of the ICOS network. A secondary goal is to demonstrate the open-source tools we developed, to be used by a multitude of stakeholders, each with their own interest in the ICOS network (see Sect. 6). We describe our methods including how station and network footprints are created and combined with relevant data in Sect. 2. The first part of the results (Sect. 3.1) focuses on the individual stations that make up the current and extended ICOS network, followed by sections for the current (Sect. 3.3) and the extended (Sect. 3.3) ICOS network. There are separate sub-sections describing what land cover types (Sect 3.2.1) and which fluxes the network represents and where important monitoring potential of these lies (Sect. 3.2.2). The discussion (Sect. 4) highlights limitations of the study and explains decisions that have influenced the results. The paper is concluded (Sect. 5) with a summary.

## 2 Methods

### 2.1 Station footprints

Footprints and modelled signals from anthropogenic emissions and the biosphere at the stations (Sect. 2.2) are computed using the ICOS Carbon Portal STILT Footprint Tool. The model set-up is described in Karstens, U., 2022 and has been used in previous studies including Levin et al., 2020, Munassar et al., 2022, and Pieber et al., 2022. The footprints are generated by STILT (Stochastic Time Inverted Lagrangian Transport; Lin et al., 2003), a Lagrangian atmospheric transport model, for the model domain 15°W-35°E and 33°N-73°N (the extent of Fig. 1). The footprints are presented on a 1/12×1/8 degrees grid with calculated surface influence ("sensitivity") in ppm / (μmol/(m²s)). The sensitivities of the cells represent the station-specific atmospheric tracer dry mole fraction dependence on fluxes and are based on the dispersion of particles transported for ten days backward from the sampling time and location (x,y,z). Meteorological conditions drive the transport and are represented by three-hourly operational ECMWF-IFS analysis/forecasts at 0.25 degrees resolution. Footprints are calculated for sampling times of every three hours (0:00, 3:00, 6:00, 9:00, 12:00, 15:00, 18:00, 21:00) and backward timestep-aggregated footprints are available for download at the Carbon Portal (https://data.icos-cp.eu/portal) and may be visualized in the STILT viewer (https://stilt.icos-cp.eu/). Footprints for summer (JJA) were used for the subsequent analyses. Additionally, footprints for winter (DJF) year 2020 were used only in the analysis of individual stations exemplified with Hyltemossa (Sect. 3.1). In case of multiple inlet heights at a station, the highest level is selected because the top-level measurements have the largest footprints and are generally chosen to provide measurements for regional and global inverse modelling systems.

## 2.2 $CO_2$ fluxes and simulated signals at the stations

Time-series of modelled $CO_2$ dry mole fraction and its components (i.e. individual anthropogenic- and biosphere-flux derived signals) are computed during the footprint calculation in the previously mentioned STILT Footprint Tool. Footprints for hourly backward time-steps are combined with temporally resolved emissions and biogenic fluxes estimated in μmol/(m²s) as described in Lin et al., 2003 (see equation 7 there).

The EDGARv4.3.2 inventory (Janssens-Maenhout et al., 2019; Gerbig and Koch, 2021b) is the basis for the anthropogenic $CO_2$ emissions. The VPRM biosphere model (Mahadevan et al., 2008; Gerbig, 2021a) is used for the biogenic fluxes. The biosphere-flux derived signals computed during the footprint calculation are attributed to groups of aggregated SYNMAP (Jung et al., 2006) land cover categories, which is also the map used to parameterize the VPRM model. The aggregation used within the STILT Footprint Tool result in broad categories such as "crop and tree", which cover over half of the land area in the model domain. To again disaggregate the land cover, we alternatively re-create the timeseries of biogenic signals by combining the footprints for hourly backward time-steps with the temporally resolved flux maps but attribute the resulting biogenic signals to the different land cover categories in HILDA (Winkler et al., 2020). The HILDA land cover is a synthesis product built on multiple heterogenous datasets including several satellite data products which were published after SYNMAP was created (Winkler et al., 2021). Another advantage is that the HILDA map used in this study represents year 2018 as opposed to the year 2000.

Ocean fluxes are currently not part of the STILT model implementation at the Carbon Portal. For this exchange, fluxes from Carbon Tracker Europe-High Resolution (van der Woude et al., 2022a; van der Woude, 2022b) were combined with the footprints for hourly backward time-steps to create timeseries of estimated ocean signals at the stations.

Whereas we focus on the surface fluxes and derived signals at the stations, total modelled mole fractions can be estimated by including "background" mole fractions to account for the contributions from global fluxes. In the Carbon Portal STILT model implementation these are taken from the Jena CarboScope globally analysed atmospheric $CO_2$ fields (Rödenbeck and Heimann, 2021). The total modelled concentrations can in turn be compared to measured concentrations to assess the performance of the modelling system; with an average correlation coefficient of 0.82 for all stations in the current (November 2022) network year 2020 there is generally a good agreement. The statistics of model vs. observation comparison for the individual stations can be found in Table A1 (Appendix A).

**2.3 Station view of land cover**

Summertime (JJA) and wintertime (JFD) average station footprints are combined with the HILDA land cover map to provide information of what land cover types are found within the footprint in different directions of the stations (see Fig. 2b). Attribution to land cover shares within individual countries is also possible using fractional country masks.

**2.4 Network footprints**

Two networks of stations are considered in this study, the 2022 ICOS atmospheric station network (see Fig. 4) and an extended ICOS atmospheric station network (see Fig. 9) which includes 19 stations that are expected to join the network in the next few years (see Fig.1). The hourly backward time-steps footprints for the individual stations in the network are reduced to grid cells with the highest sensitivity values that in combination add up to 50% of the sum of the hourly backward time-step footprints

sensitivities. This follows the approaches of Henne et al. (2010) and Oney et al. (2015) and is intended to emphasize areas with significant local influence. The hourly 50% footprints of the individual stations are combined in final hourly backward network footprints where in the case of multiple stations with sensitivity to the same footprint grid cell, the maximum cell value is used. The hourly backward time-step network footprints associated with receptor measurements every three hours can in turn be combined with underlying land cover as fluxes for subsequent analyses (see Sect. 2.5).

To estimate the overlap in sensitivities between a current network footprint and the 50% footprint of a station that is included in the extended network, the effect of its inclusion in an updated network footprint is analysed; the difference between the spatial sums of the two network footprint sensitivities is compared to the spatial sum of the 50% station footprint. If there is no overlap between the current network and the footprint of the station joining, the difference between the two network

footprints is the same as the sum of the (50%) station footprint.

**2.5 Network views of land cover and associated fluxes**

Average summertime network footprints are combined with the HILDA land cover map to analyse what land cover types are sensed in individual countries and in Europe as a whole (referred to as "LC-view"). Fluxes associated with the different land cover types are established from the combination of hourly backward time-steps network footprints with temporally

corresponding flux maps (referred to as "GEE-view") and averaged for the summer. We use GEE (Gross Ecosystem Exchange) rather than NEE (Net Ecosystem Exchange = GEE + Respiration) as footprint-weight to prevent nearly cancelling photosynthesis and respiration signals to influence the network view. The GEE-view thus highlights areas with high biogenic activity which are especially important to monitor; high activity generally means greater uncertainties in the current estimates and higher potential for long-term carbon storage. The network views are evaluated here exclusively for the summer when

GEE is highest but using other time periods have proven to give similar results (see Sect. 4).

The LC- and GEE-views of the network are compared to what "equal views" would yield. An equal view lens is established from corresponding hourly backward network footprints where the mean sensitivity given a chosen region is distributed to all footprint cells (see Eq. 2). These are used to establish alternative LC- and GEE- views which are used as baseline to establish relative over- or underrepresentation. It should be noted that to have an "equal view" of the studied region is not always desirable as some fluxes might be more relevant to monitor than others. Consequently, over- and underrepresentations are not inherently negative for the network (see Eq. 3). Note that an equal LC-view reflects the relative distribution of area associated with different land cover types. In terms of the GEE-view, variations in the underlying biogenic fluxes of each land cover mean that there can be over- and underrepresentation even if the relative shares are the same (see Fig. 5 and Fig. 8).

The following definitions are given to help the reader:

We consider the model domain 15°W to 35°E and 33°N-73°N:

For a given country or region, $C$, we can look at $C_{i,j}$ which is the fraction of the country/region in a given grid cell $(i, j)$. We consider specific land cover types, $LC$, and use $LC_{i,j}$ which is the fraction of land cover within a given grid cell.

We establish the network view ($N_{i,j}(T)$) and equal view ($NEQ_{i,j}(T)$) of the flux land scape ($GEE_{i,j}(t_k)$): For each grid cell $i, j$ and hour $t_k$ leading up to when the air arrives at the receptor (T) where hour $t_k = t_1, t_2, t_3, \ldots, t_{240}$ (here $t_{240}$ is T, the time the air arrives at the receptors of the stations in the network).

$$N_{i,j}(T) = C_{i,j} \cdot LC_{i,j} \sum_{k=1}^{240} NFP_{i,j}(t_k) \cdot \left| GEE_{i,j}(t_k) \right|$$

(1)

$$NEQ_{i,j}(T) = C_{i,j} \cdot LC_{i,j} \sum_{k=1}^{240} \frac{\sum_{m,n} C_{m,n} \cdot NFP_{m,n}(t_k)}{\sum_{m,n} C_{m,n}} \cdot \left| GEE_{i,j}(t_k) \right|$$

(2)

Where $C$ and $LC$ are the fractional country grid of selected country or region. $NFP$ is the network footprint (see Sect. 2.4) and $GEE$ is the flux map and these change with time ($t_k$). $m,n$ are sum indices that run over all cell coordinates in the model grid.

The relative flux representation ($REP(T)$, used in Fig. 5b and Fig. 8b is the ratio between the total sensing within the grid cells of the network view ($N_{i,j}(T)$) and the equal view $NEQ_{i,j}(T)$).

$$REP(T) = \frac{\sum_{i,j} N_{i,j}(T)}{\sum_{i,j} NEQ_{i,j}(T)} \qquad (3)$$

The relative monitoring potential maps, *MP(T)*, show the difference in the sensing between the network view and the equal view within the individual grid cells of the model.

$$MP(T) = max\left( NEQ_{i,j}(T) - N_{i,j}(T), 0 \right) \qquad (4)$$

Cells where the equal view ($NEQ_{i,j}$) is greater (more uptake), than the network view ($N_{i,j}$) will have positive values in the
relative monitoring potential map (*MP*) and all other cells have the value zero. Monitoring potential becomes especially high in areas where the current network is relatively blind, and the activity of the specific flux is relatively high. For the monitoring potential maps for the extended network, the equal view ($NEQ_{i,j}$) is kept the same as for the current network to facilitate effective comparison between the maps.

**3 Results**

We focus the first part of the results (Sect. 3.1) on characteristics of stations in the ICOS current and extended networks and demonstrate our capacity for a deeper station analysis for the Swedish station Hyltemossa (HTM150). We then quantify the monitoring network views in its current (Sect. 3.2) and extended (Sect. 3.3) ICOS configuration, with a focus on the new concept of "relative monitoring potential".

**3.1 The view from individual stations**

| Station | Crop | Coniferous forest | Mixed forest | Pasture | Broadleaf forest | Urban | Grass & shrub | Other land cover | Ocean | Energy | Transport | Industry | Residential |
|---------|------|------|------|------|------|------|------|------|------|------|------|------|------|
| ARN100* | -0.73 | -0.27 | -0.89 | -0.46 | -0.42 | -0.24 | -0.23 | -0.03 | -0.01 | 0.35 | 0.36 | 0.23 | 0.05 |
| BIK300* | **-4.41** | -2.22 | -1.59 | -1.71 | -1.13 | -0.68 | **-1.19** | -0.11 | -0.03 | 0.46 | 0.31 | 0.20 | 0.09 |
| BIR075 | -1.20 | -4.15 | -0.73 | -0.50 | -0.29 | -0.25 | -0.50 | -0.15 | -0.12 | 0.56 | 0.29 | 0.19 | 0.06 |
| CMN760 | -2.28 | -1.12 | -1.56 | -1.36 | **-3.78** | -0.84 | -0.73 | -0.10 | -0.01 | 0.39 | 0.43 | 0.28 | 0.11 |
| EST110* | -2.58 | -2.73 | **-3.21** | -0.78 | -1.56 | -0.32 | **-1.47** | **-0.28** | -0.07 | 0.43 | 0.17 | 0.15 | 0.05 |
| FKL015* | -2.72 | -0.70 | -0.90 | -1.11 | -1.53 | -0.44 | -0.63 | -0.07 | 0.13 | 0.52 | 0.38 | 0.30 | 0.10 |
| GAT344 | -3.63 | -1.73 | -1.02 | -1.28 | -0.49 | -0.76 | -0.47 | -0.11 | -0.07 | 0.69 | 0.45 | 0.39 | 0.16 |
| HAC* | -3.01 | -0.97 | -1.31 | -1.82 | **-2.22** | -0.54 | -0.87 | -0.08 | 0.01 | 0.37 | 0.28 | 0.22 | 0.07 |
| HEL110 | -2.27 | -0.99 | -0.77 | -1.14 | -0.34 | -0.48 | -0.36 | -0.11 | -0.09 | 0.65 | 0.39 | 0.27 | 0.12 |
| HPB131 | -2.80 | -1.66 | **-2.79** | **-4.27** | -0.93 | **-1.58** | -0.45 | -0.19 | -0.03 | 0.50 | 0.69 | 0.41 | 0.23 |

| | | | | | | | | | | | | |
|---|---|---|---|---|---|---|---|---|---|---|---|---|
| HTM150 | -2.71 | -2.53 | -1.46 | -0.67 | -0.77 | -0.44 | -0.49 | -0.14 | -0.10 | 0.46 | 0.32 | 0.23 | 0.07 |
| HUN115* | **-6.02** | -1.95 | **-3.11** | -1.59 | **-2.30** | **-1.58** | -1.07 | -0.10 | -0.03 | 0.67 | 0.68 | 0.47 | 0.18 |
| IPR100 | -3.74 | -1.39 | -2.02 | **-1.95** | **-4.15** | **-2.43** | -0.71 | **-0.92** | -0.01 | 0.85 | **1.61** | **1.74** | **0.55** |
| JFJ | -1.81 | -1.10 | -1.95 | -1.86 | -0.91 | -0.88 | -0.47 | -0.21 | -0.02 | 0.26 | 0.38 | 0.23 | 0.12 |
| JUE120 | -2.97 | -0.84 | -1.12 | -1.33 | -0.53 | **-1.45** | -0.24 | -0.09 | -0.07 | **5.60** | **1.15** | **1.33** | **0.46** |
| KAS* | -4.05 | -2.25 | -1.83 | **-2.04** | **-1.61** | -1.18 | -0.93 | -0.10 | -0.02 | 0.61 | 0.37 | 0.32 | 0.12 |
| KIT200 | -3.43 | -1.06 | -1.77 | -1.25 | -1.24 | -1.31 | -0.32 | -0.05 | -0.05 | **1.63** | **0.83** | **0.67** | **0.33** |
| KRE250 | **-4.94** | -2.25 | -1.57 | -1.56 | -0.82 | -1.10 | -0.51 | -0.08 | -0.04 | 0.74 | 0.53 | 0.41 | 0.16 |
| LAH032* | -1.90 | **-4.55** | **-2.75** | -0.66 | -0.92 | -0.34 | **-1.13** | -0.25 | -0.11 | 0.71 | 0.23 | 0.38 | 0.06 |
| LIN099 | -3.71 | -2.50 | -1.22 | -1.21 | -0.62 | -0.87 | -0.62 | -0.14 | -0.06 | **1.53** | 0.62 | **0.71** | **0.24** |
| LMP | -1.63 | -0.60 | -0.93 | -0.85 | -1.03 | -0.44 | -0.46 | -0.05 | 0.26 | 0.33 | 0.40 | 0.20 | 0.07 |
| LMU080* | -1.40 | -0.51 | -1.20 | -0.68 | -1.07 | -0.29 | -0.28 | -0.03 | -0.03 | 0.30 | 0.36 | 0.25 | 0.07 |
| LUT | -3.11 | -0.95 | -0.78 | -1.58 | -0.35 | -0.66 | -0.37 | -0.22 | -0.09 | **1.35** | 0.58 | 0.42 | 0.20 |
| MAH* | -0.87 | -0.62 | -0.50 | -1.39 | -0.16 | -0.22 | -0.28 | -0.08 | **-0.23** | 0.25 | 0.25 | 0.12 | 0.06 |
| MAJ100* | -0.70 | -0.37 | -1.15 | -0.72 | -0.43 | -0.24 | -0.29 | -0.02 | -0.02 | 0.12 | 0.23 | 0.14 | 0.04 |
| MHD* | -0.70 | -0.41 | -0.32 | -1.08 | -0.09 | -0.16 | -0.43 | -0.15 | **-0.16** | 0.17 | 0.19 | 0.09 | 0.05 |
| NOR100 | -1.72 | **-5.09** | -1.53 | -0.35 | -0.25 | -0.23 | -0.48 | **-0.31** | -0.12 | 0.39 | 0.21 | 0.24 | 0.04 |
| OPE120 | -3.20 | -0.67 | -0.96 | -0.99 | -1.18 | -0.70 | -0.23 | -0.05 | -0.05 | 0.45 | 0.48 | 0.36 | 0.14 |
| OXK163 | -3.55 | -2.05 | -1.49 | -1.52 | -0.78 | -1.18 | -0.39 | -0.06 | -0.05 | 0.78 | 0.61 | 0.41 | 0.20 |
| PAL | -0.37 | -4.49 | -0.58 | -0.14 | -0.12 | -0.07 | **-1.43** | -0.22 | -0.10 | 0.09 | 0.06 | 0.05 | 0.01 |
| PCW150* | **-5.02** | -2.05 | -1.36 | -1.55 | -0.81 | -0.85 | -0.91 | -0.12 | -0.04 | 0.96 | 0.43 | 0.30 | 0.15 |
| PRS | -1.54 | -1.10 | -1.57 | -1.77 | -0.96 | -0.75 | -0.36 | -0.27 | -0.02 | 0.24 | 0.34 | 0.20 | 0.10 |
| PSE150* | **-5.10** | -2.47 | -1.84 | **-1.91** | -1.38 | **-1.38** | **-1.19** | -0.09 | -0.03 | 0.88 | 0.47 | 0.59 | 0.16 |
| PUI084 | -0.86 | **-5.06** | -1.98 | -0.21 | -0.26 | -0.15 | -0.53 | **-0.56** | -0.07 | 0.53 | 0.12 | 0.11 | 0.04 |
| PUY | -1.55 | -0.65 | -1.53 | -1.74 | -1.09 | -0.41 | -0.29 | -0.04 | -0.03 | 0.20 | 0.27 | 0.17 | 0.07 |
| RGL090* | -2.13 | -0.56 | -0.74 | -1.85 | -0.17 | -0.55 | -0.27 | -0.07 | -0.12 | 0.60 | 0.60 | 0.31 | 0.18 |
| SAC100 | -2.49 | -0.50 | -0.64 | -0.95 | -0.66 | -0.68 | -0.21 | -0.05 | -0.07 | 0.48 | **1.07** | **1.04** | **0.45** |
| SMR125 | -1.23 | **-5.57** | -1.62 | -0.25 | -0.31 | -0.19 | -0.52 | **-0.42** | -0.09 | 0.34 | 0.16 | 0.17 | 0.05 |
| SNZ* | -3.92 | -2.10 | -1.44 | -1.43 | -0.77 | -0.91 | -0.60 | -0.09 | -0.04 | 0.73 | 0.43 | 0.33 | 0.14 |
| SSL* | -3.10 | -1.33 | -2.04 | -1.51 | -1.35 | -1.34 | -0.31 | -0.06 | -0.04 | 0.46 | 0.58 | 0.40 | 0.19 |
| STE252 | -3.56 | -1.08 | -0.90 | -1.71 | -0.42 | -0.97 | -0.39 | -0.11 | -0.07 | **0.98** | 0.56 | 0.52 | 0.22 |
| SVB150 | -0.64 | **-5.92** | -0.94 | -0.22 | -0.18 | -0.12 | -0.75 | -0.24 | -0.10 | 0.16 | 0.10 | 0.08 | 0.02 |
| TOH147 | -3.57 | -1.90 | -1.13 | -1.24 | -0.80 | -1.03 | -0.37 | -0.07 | -0.06 | 0.87 | 0.63 | 0.63 | 0.21 |
| TRN180 | -2.66 | -0.48 | -0.72 | -0.90 | -0.77 | -0.62 | -0.22 | -0.04 | -0.06 | 0.28 | 0.49 | 0.32 | 0.16 |
| UTO | -1.31 | -2.96 | -1.32 | -0.39 | -0.43 | -0.21 | -0.55 | -0.19 | **-0.24** | 0.50 | 0.19 | 0.24 | 0.04 |
| VSD006* | -1.68 | -2.76 | -1.84 | -0.57 | -0.69 | -0.27 | -0.81 | -0.18 | **-0.29** | 0.39 | 0.24 | 0.19 | 0.05 |
| VTO014* | -0.60 | -0.34 | -0.32 | -1.01 | -0.09 | -0.17 | -0.26 | -0.06 | **-0.17** | 0.19 | 0.20 | 0.09 | 0.05 |
| WAO | -4.08 | -0.75 | -0.75 | -1.18 | -0.23 | -0.64 | -0.35 | -0.17 | -0.09 | 0.82 | **0.82** | 0.40 | 0.18 |

| | | | | | | | | | | | | | |
|---|---|---|---|---|---|---|---|---|---|---|---|---|---|
| WES | -2.30 | -1.04 | -0.78 | -0.99 | -0.31 | -0.44 | -0.40 | -0.12 | -0.09 | 0.97 | 0.47 | 0.22 | 0.12 |
| ZSF | -2.19 | -1.75 | **-2.07** | **-2.37** | -0.92 | -1.10 | -0.52 | -0.18 | -0.03 | 0.34 | 0.47 | 0.30 | 0.15 |
| **Average** | -2.55 | -1.86 | -1.37 | -1.23 | -0.89 | -0.69 | -0.56 | -0.15 | -0.06 | 0.66 | 0.45 | 0.36 | 0.14 |
| **Std** | 1.34 | 1.47 | 0.66 | 0.70 | 0.81 | 0.49 | 0.32 | 0.15 | 0.08 | 0.78 | 0.28 | 0.30 | 0.11 |


**Table 1: Summertime (JJA) average land cover GEE, anthropogenic, and ocean signals (in ppm) at the stations in the current (November 2022) and extended (\*) ICOS atmospheric network, limited to stations within the model domain. For each category, the values of the five stations with highest signals are in bold. See Table B1 (Appendix B) for more information about the stations and the table provided as supplemental material for respiration and total CO$_2$ signals.**

The individual stations of our studied networks show large variations in expected signals, with most with stronger signals from biogenic fluxes than anthropogenic emissions in the summer of year 2020 (Table 1). Considering the sensitivity to biogenic activity, the highest average (negative) signal is associated with cropland, followed by coniferous forest. Stations in countries with large areas associated with cropland, such as Czech Krésin (KRE250) and Dutch Lutjewad (LUT), show some of the largest signals associated with cropland. Similarly, coniferous forest is found in abundance in northern Europe which means

large signals at Swedish Norunda (NOR100) and Svartberget (SVB150), and Finish SMEAR (SMR125) and Puijo (PUI084). Although mole fraction contributions typically reflect land cover fractions within the footprints, their ratios can vary over space and time due to variations in the underlying vegetation fluxes of each land cover type. For example, forests have normally higher photosynthetic activity than cropland which calls for the distinction between LC-view and GEE-view.

Signals from anthropogenic emissions are generally expected to be small because ICOS targets natural fluxes. For several stations, this targeting is proving successful and about one third of the stations have average anthropogenic signals below one ppm for the summer in year 2020. These include "background stations" with limited influence from local surface fluxes thanks to strategic placement by the ocean, including Lampedusa (LMP), Mace Head (MHD) and Malin Head (MAH), or on remote mountain tops, such as Jungfraujoch (JFJ), Plateau Rosa (PRS) and Puy de Dôme (PUY). However, high emission intensity in

central European countries makes it hard to avoid significant emission sources within the large footprints of atmospheric stations. On average, emissions related to energy production cause the largest signals, especially at German and Polish stations. However, it is important to remember that the signal averages include peaks in anthropogenic signals during particular hours especially when the wind transports air from large point source emitters. For example, the German station Jülich (JUE) is located only 10 km from a coal-fired power plant that accounts for about 4% of Germany's total emissions (E-PRTR, 2020).

The average signal is 5.6 ppm but below 1.0 ppm about 20% of the time. Careful sub-sampling of time series, as suggested also by Oney et al. (2015), could allow for either avoiding anthropogenic influence or concentrating on its analysis. A deeper analysis per station is facilitated by our tools, as exemplified next for the station Hyltemossa.

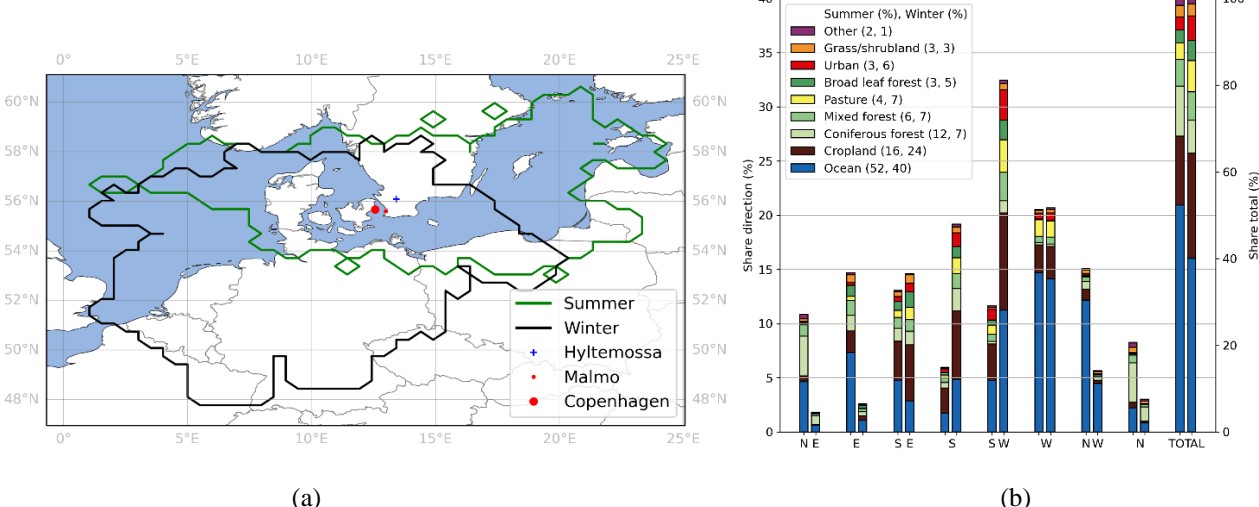

(a)                                                                          (b)

Figure 2: (a) Hyltemossa's 50 % footprint area for summer (JJA) and winter (JFD) year 2020 and (b) land cover shares weighed by the seasonal footprints split by direction.

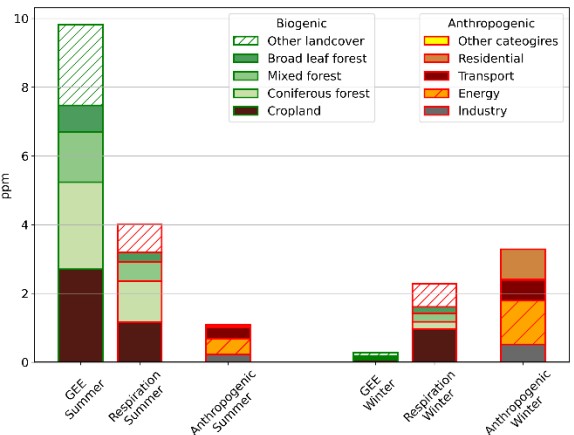

Figure 3: Biogenic and anthropogenic signals at Hyltemossa, summer (JJA) and winter (JFD) year 2020. The biogenic signals are attributed to different land cover types and the anthropogenic signals by source categories.

Hyltemossa is located in southern Sweden in an area of managed coniferous forests. Footprints calculated for the 150 meter inlet height are analysed, with major cities such as Malmo and Copenhagen well-within the large general footprint region (see Fig. 2a). The anthropogenic signals are nevertheless small, and it is an appropriate station for large-scale monitoring of oceanic and biospheric fluxes. Seasonal differences in footprints are substantial with larger footprints in the winter, extending south-west to an area characterized by more cropland and into densely populated regions such as Northern Germany, and the Netherlands. The size and extent of the average footprints for summer and winter can be further examined in Fig. 2a complemented by division of footprint sensitivity by land cover in Fig. 2b. The summer footprint extends far east and west of the station, which explains the large share of sensitivity to ocean/sea. Cropland has the second largest share within both the

summer and winter footprints. Coniferous forest is the third most sensed land cover with a higher relative share in the summer because of favourable conditions for conifers mainly in the northern latitudes. Without analysis of the footprints, even higher

sensing of coniferous forests would be expected with abundance in the vicinity of the station and in Sweden as a whole (60%).

The summertime atmospheric $CO_2$ signal at Hyltemossa strongly reflects the biospheric fluxes from the land cover types under its footprint, but the share of the signal associated with coniferous forest is almost as large as cropland despite significantly more cropland within the footprint (see Fig. 2b and Fig. 3). The coniferous forest flux signals mostly come from within

Sweden's border (63% of signal) while cropland signals at Hyltemossa originate mainly from outside Sweden (35% of signal) including Poland (13%), Denmark (10%), and Germany (9%). During the winter, biosphere respiration of $CO_2$ is almost as large signal as the anthropogenic contribution, which is dominated by energy production, followed by residential. Interestingly, emission sources within Sweden only contribute about 23% of the anthropogenic signal at the station. The remainder is transported from emission sources further away, emphasizing the importance of long-range transport.

**3.2 The view from the current ICOS network**

We will start our analysis of the view from the current ICOS network by quantifying what land cover types are sensed by ICOS (Sect. 3.2.1). The pan-European view of the network is complemented with the view within individual countries to account for the uneven representation within Europe. A view of the fluxes associated with the different land cover types follows which is also used to produce monitoring potential maps (Sect. 3.2.2).


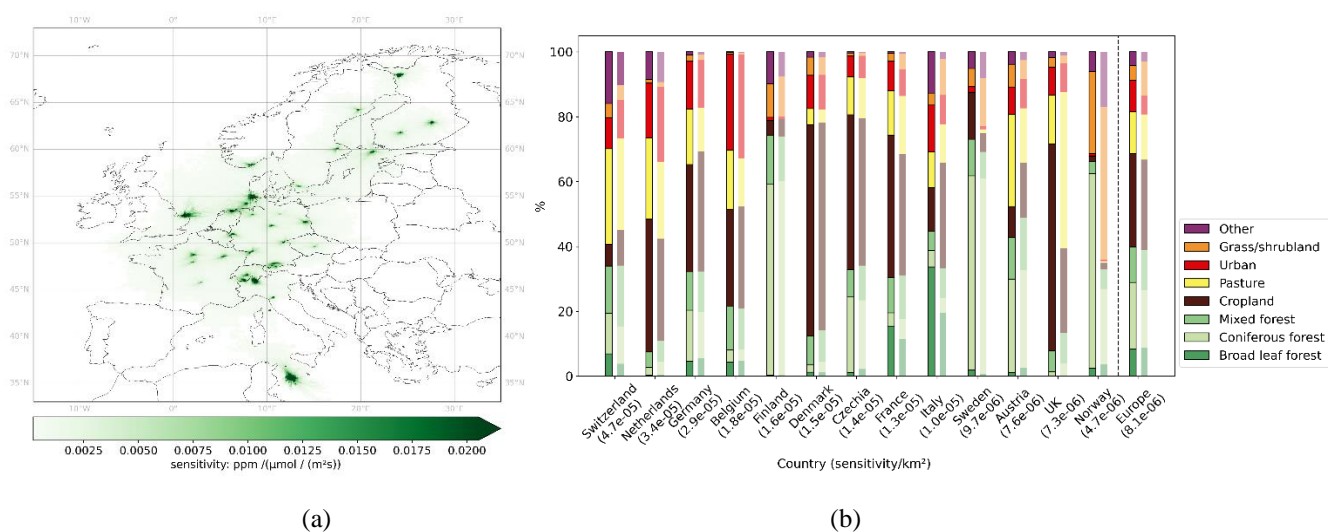

(a) (b)

**Figure 4: (a) Summer (JJA) year 2020 average network footprint. (b) Sensitivity of network land cover (left bars) compared to country shares of land cover (right bars) in ICOS members countries. The graph is sorted from highest to lowest sensitivity per km²**
**(values found in parentheses).**

### 3.2.1 Land cover types sensed by ICOS

The share of different land cover types within Europe is in general agreement with the shares of different land cover types sensed by the ICOS network, with the exceptions of an overrepresentation of coniferous forests and underrepresentation of grass & shrubland (see Fig. 4b, "Europe"). Results for the same type of analysis within the individual ICOS member countries generally show larger differences for countries only partially within the average network footprint (see Fig. 4a). Coniferous forest is predominately sensed within the Nordic countries with network shares in Finland and Sweden matching their national shares, whereas Norway's only ICOS station is in an area abundant with coniferous forests which skews the network's sensing capacity. What is mainly missing is grass & shrubland which cover almost half of Norway. Other countries with high shares of grass & shrubland including Croatia, Serbia, and Albania are barely within the network view and contribute to the relative underrepresentation on European scale. Broadleaf forests are fairly well represented on European scale but are relatively overrepresented in many of the ICOS member countries including Italy, France, and Switzerland. Countries outside the network view with extensive areas of broadleaf forests, such as Slovakia and Slovenia, balance the local overrepresentation in the European scale consideration. This stresses the need for network-assessment on multiple scales, and for formal quantification to be complemented by a visual inspection of the network.

### 3.2.2 Fluxes sensed by ICOS

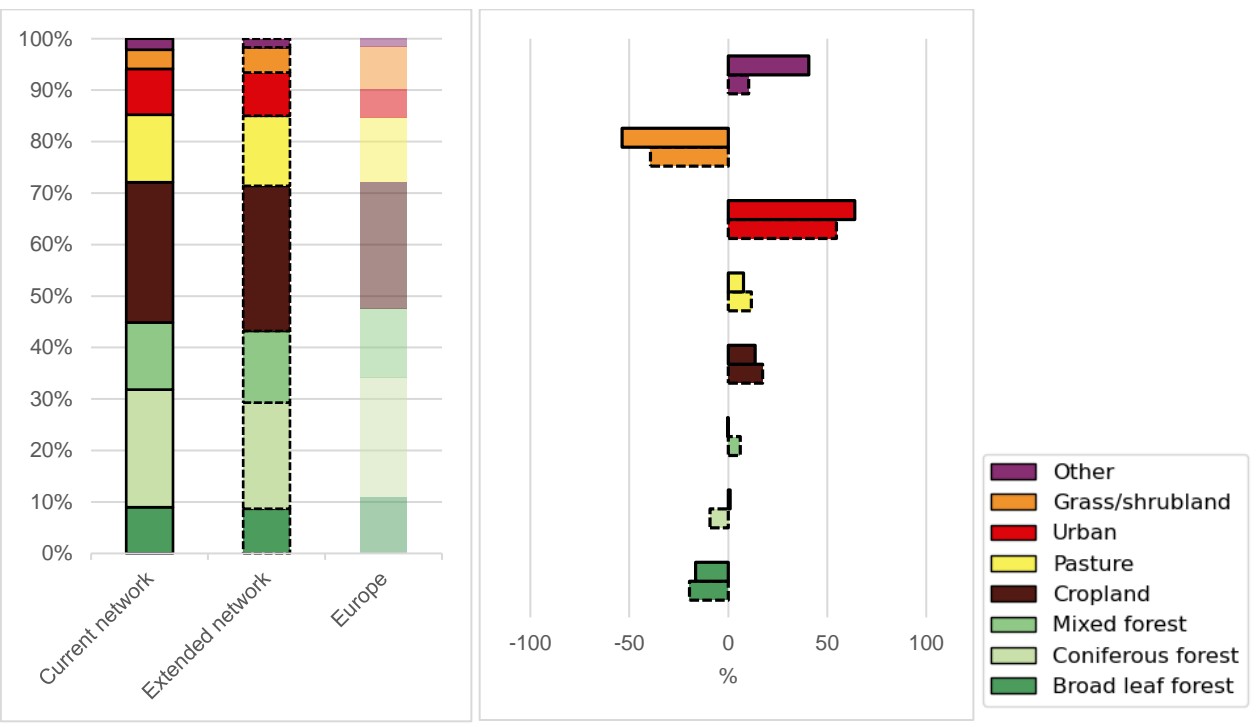

(a)                                                                              (b)

**Figure 5: (a) Share of flux (GEE) per land cover within Europe compared to the network GEE-view for the current and extended**
**ICOS network within Europe. (b) The over- (+) or under (-) representation of the current ICOS network (upper bars) compared to the extended (lower bars) ICOS network within Europe (see Sect. 2.5).**

When the GEE-view of the network within Europe is considered, relative underrepresentation of broadleaf forest fluxes, in addition to grass & shrubland fluxes, is revealed (see Fig. 5a). The underrepresentation of broadleaf forest fluxes (see Fig. 5b), despite a fair LC-view (see Fig. 4b), means that broadleaf forests outside the focus of the network are more active than those
currently sensed. These active, relatively unmonitored forests are highlighted in our monitoring potential map (Fig. 6a) that indicates areas outside the reach of the current network as high in potential, especially in south-eastern Europe. Within the current ICOS member countries, Italy shows high potential for broadleaf forest flux monitoring south of Monte Cimone (CMN) and, to lesser degrees, areas in southern France and eastern Czech Republic. Grass & shrubland is the most underrepresented flux (see Fig. 5b) and has the greatest potential for monitoring in Serbia and Croatia (see Fig. 6b). Within the ICOS member
countries, Scandinavia has the highest potential (see Fig. 6b).

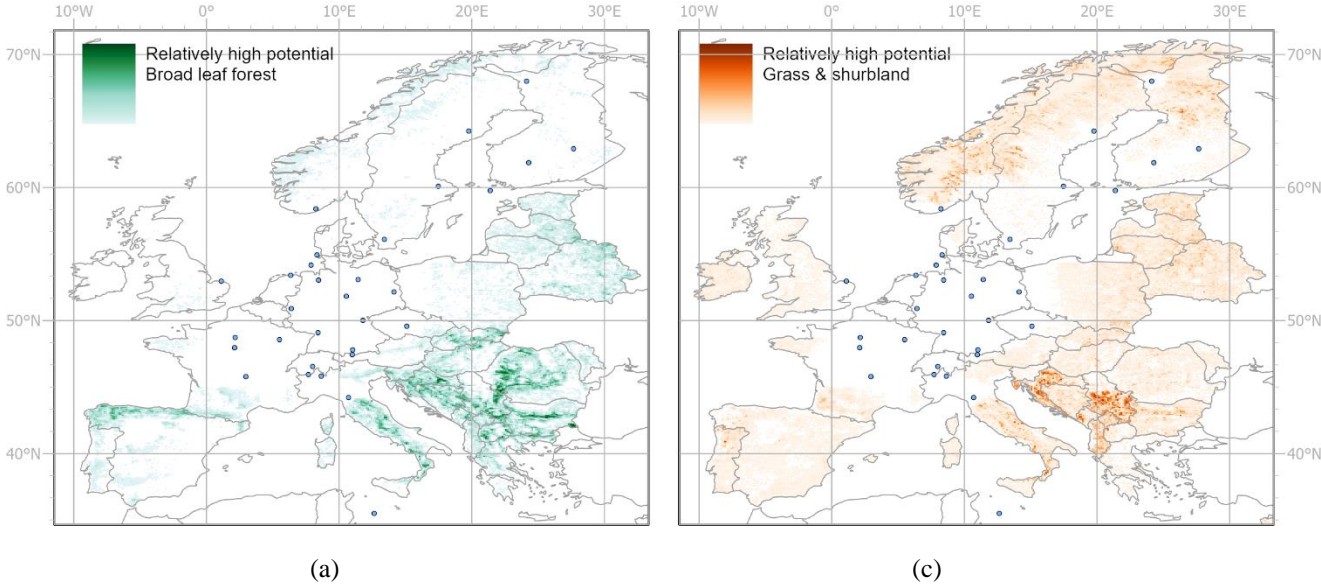

(a)                                                                              (c)

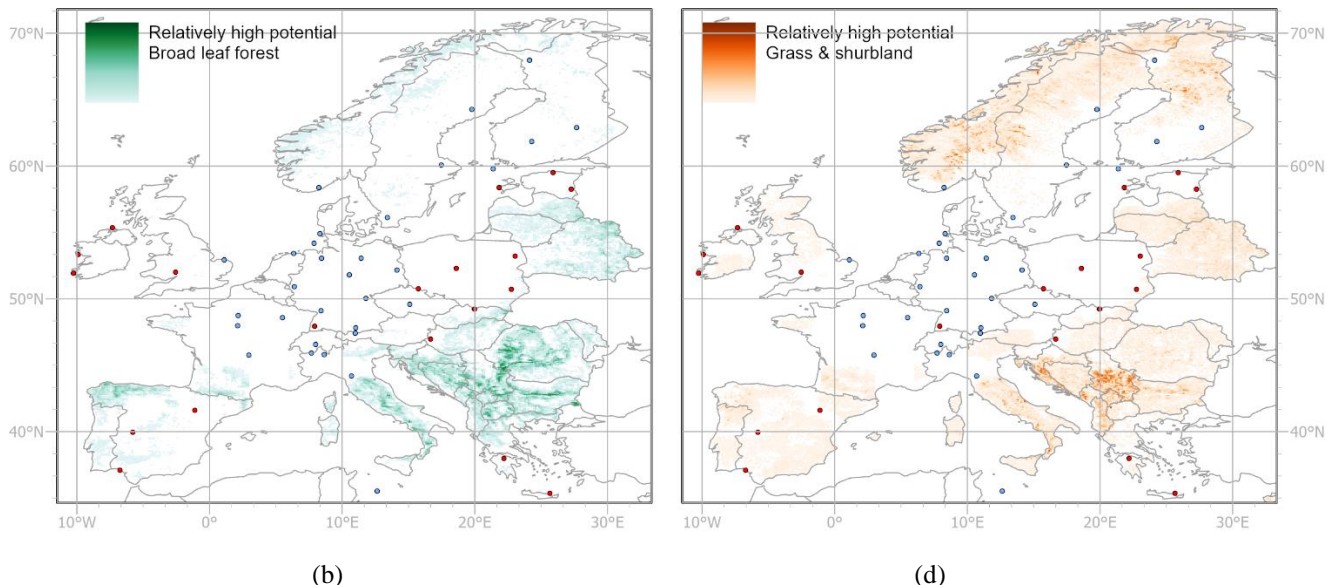

(b)                                      (d)

**Figure 6: European scale relative monitoring potential of (a) broadleaf forest fluxes (GEE) for the current and (b) extended ICOS networks and (c) grass & shrubland fluxes (GEE) for the current and (d) extended ICOS network.**

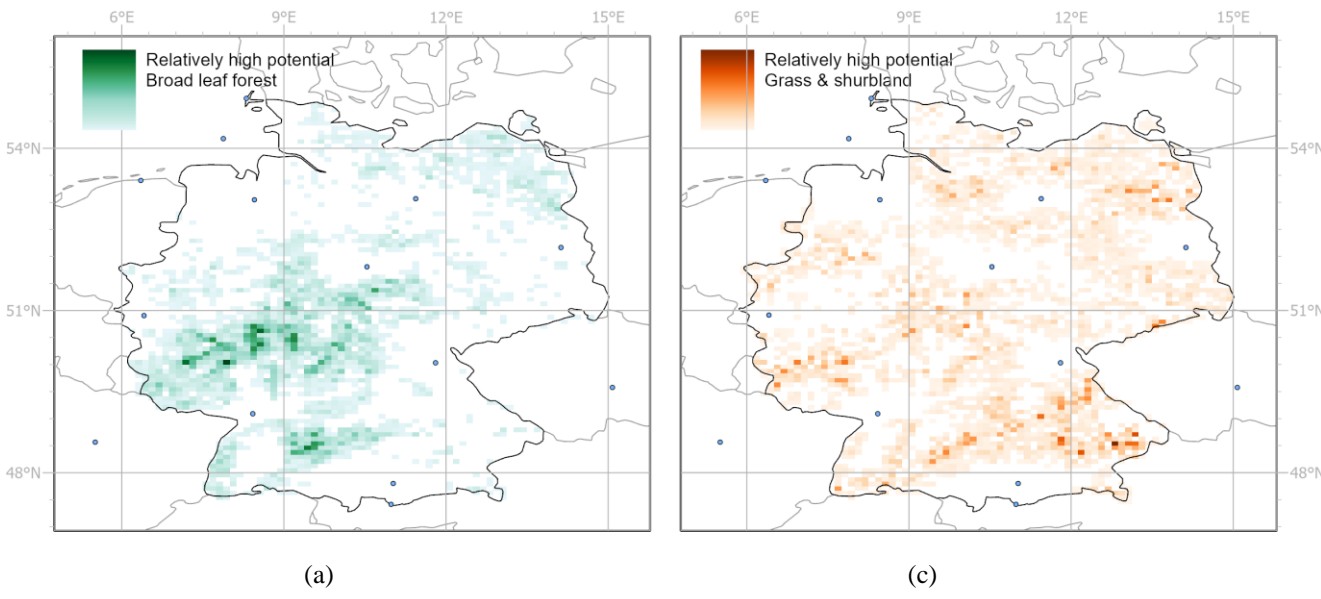

(a)                                      (c)

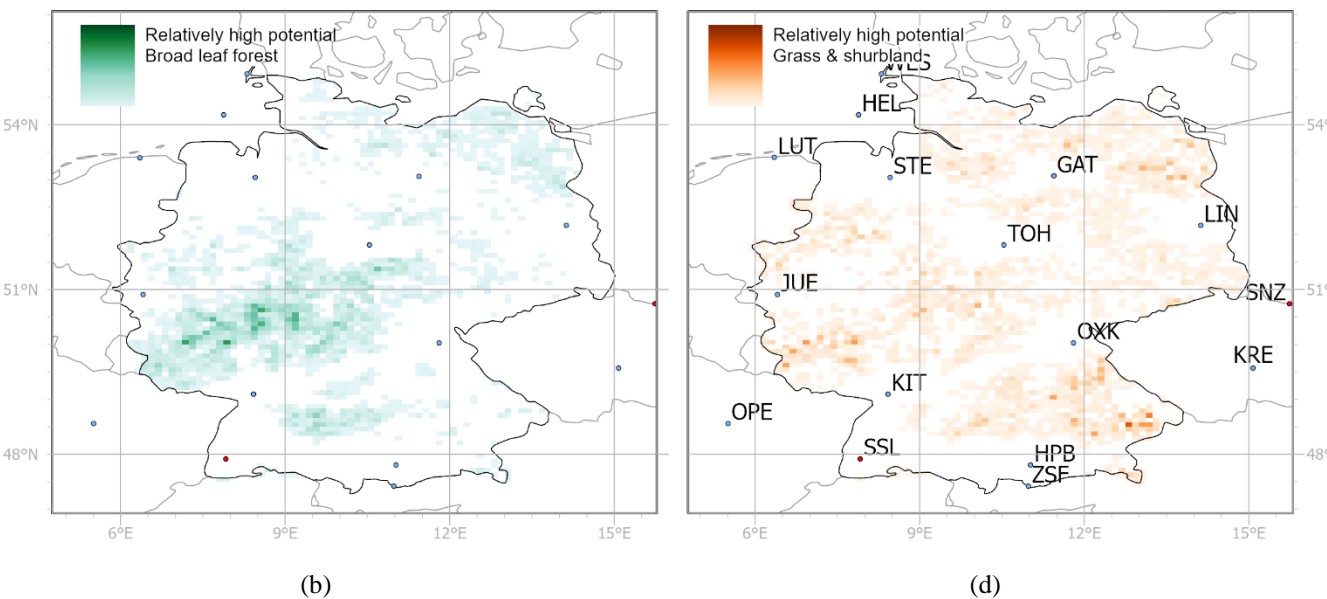

(b)                                   (d)

**Figure 7: Same as Fig. 6, but relative monitoring potential within Germany as opposed to Europe. In the European context, Germany is well-monitored and appears to have no relative monitoring potential (white).**

The overrepresentation of fluxes associated with the land cover type "urban" (see Fig. 5b), despite the ICOS network targeting

natural fluxes, is explained by the relatively high density of stations in western and central Europe; countries with highest sensitivity per area unit (Switzerland, the Netherlands, and Germany; see Fig. 4b) are also counties with some of the highest population densities. The high sensitivity of the network within these countries compared to the rest of Europe also means that their fluxes are relatively well-monitored and appear low in monitoring potential when all of Europe is considered (see Eq. 1-Eq. 4). This should not be interpreted as if their monitoring is "complete" and for expansions of national networks it is advisable

to consider the relative sensing within the individual countries which we will illustrate for Germany.

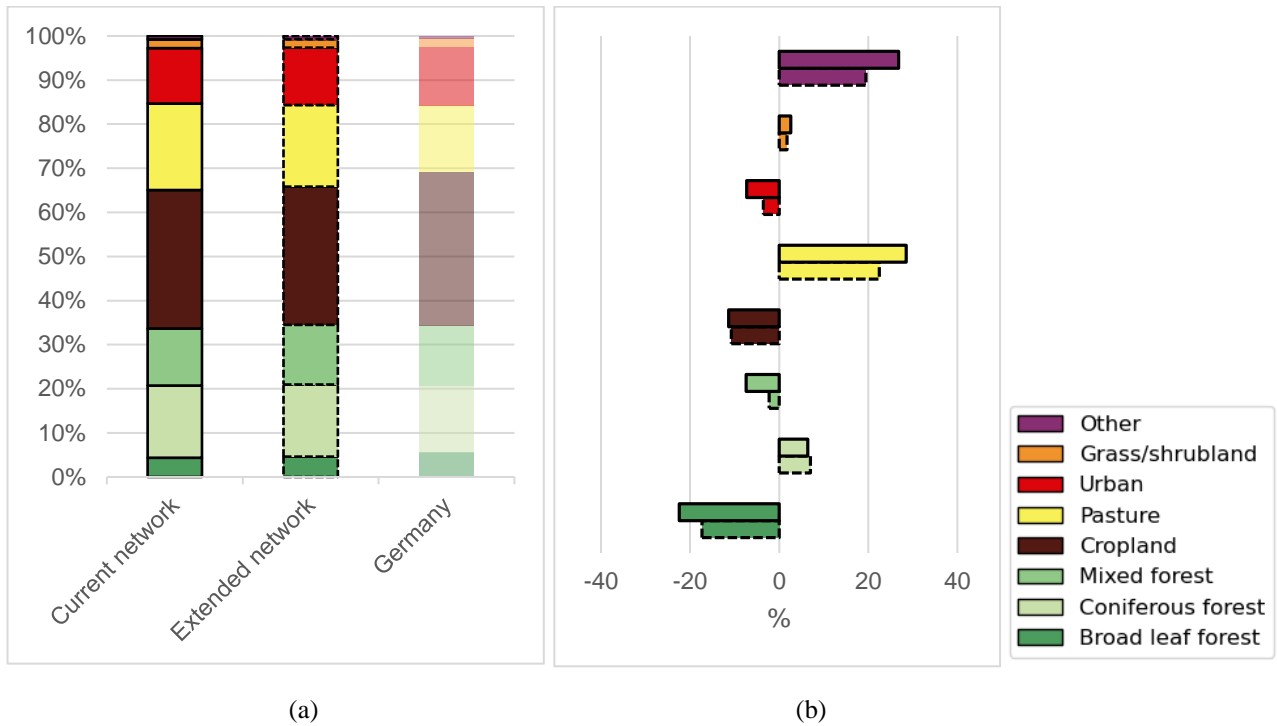

(a)                                                         (b)

**Figure 8: (a) Share of flux (GEE) per land cover type within Germany compared to the network GEE-view of Germany for the current and extended ICOS network (b) The over- or under- representation of the current ICOS network (upper bars) compared to the extended (lower bars) network within Germany (see Sect. 2.5).**

Germany is the ICOS member country with the largest number of atmospheric stations (ten) and has the third highest sensing capacity per area unit in Europe (see Fig. 4b). Although the stations are spread throughout the country, the network represents some land cover fluxes better than other; broadleaf forest fluxes are relatively underrepresented also within Germany (Fig. 8b), with high monitoring potential in the central part of the country extending south to Karlsruhe (KIT) and west to Jülich (JUE) (see Fig. 7a). However, broadleaf forests are only associated with 5.5% of the biogenic flux in Germany compared to 14% for mixed forests (see Fig. 4b) which is also underrepresented but to a lesser degree (see Fig. 8b). Areas of high monitoring potential of the two forest types show a great deal of overlap but with mixed forests additionally having high potential in the south-eastern part of the country (not shown). The use of monitoring potential maps is not limited to underrepresented fluxes as demonstrated for relatively well-monitored grass & shrubland within Germany (see Fig. 7c). This is important because a certain flux type might be targeted, and overrepresentation is in those cases desirable.

### 3.3 The view from the extended ICOS network

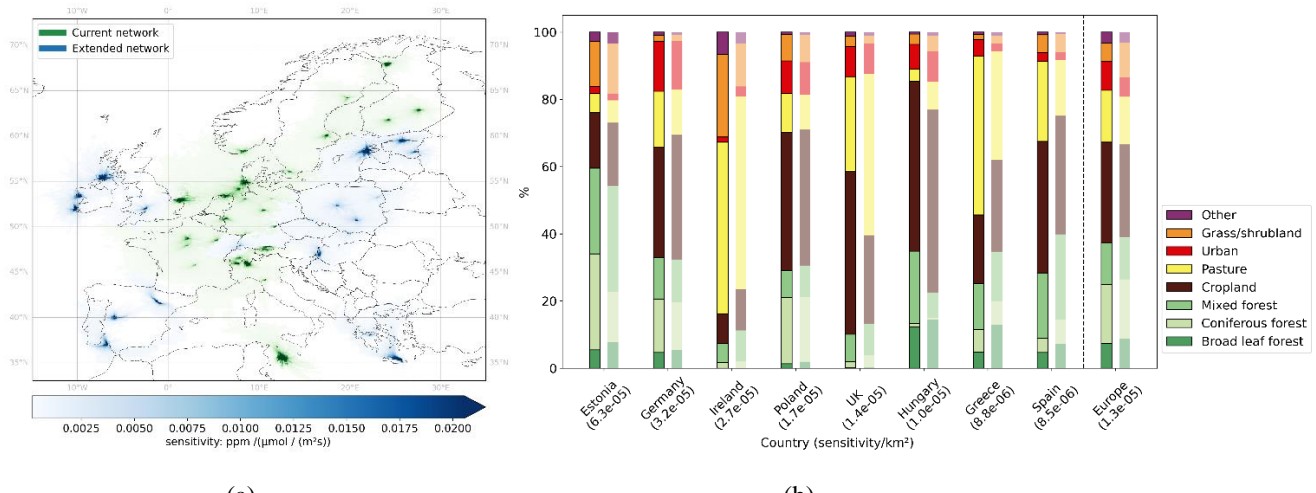

(a)                                                   (b)

**Figure 9: (a) Summer (JJA) 2020 extended network footprint overlaid on the current network footprint. The same level of color saturation of green and blue have the same meaning (b) Sensing of network to land cover (left bars) compared to country shares of land cover (right bars) in countries with stations added.**

The ICOS network is continuously expanding, and an additional 19 stations are expected to join in the next few years. This will greatly extend the reach of the network, especially as the planned stations are mainly sensitive to areas outside the focus of the current network (see Fig. 9a). Overlap (see Sect. 2.4) with the sensing capacity of the current network is close to zero for Spanish and Irish stations with respect to their 50% summertime footprints, and well-below 10% for all other added stations with exceptions for German Schauinsland (SSL) and the Polish mountain-station Sněžka (SNZ). However, Schauinsland is added to a national network of ten other stations and the overlap is only 21%. This demonstrates the potential to fill network monitoring gaps also in relatively well-monitored countries. Whether the added stations make for a network with more equal representation of different land cover, and land cover associated fluxes depends on the scale of the analysis; a fairer representation of land cover is evident especially within countries previously without stations, whereas the representation on European scale shows only small changes (see Fig. 9b). The differences in shares for Europe indicate that stations are added in areas of more cropland (particularly the added Hungarian station and Polish stations, see Table 1) and pastures (particularly Irish, UK and Greek stations, see Table 1) than in coniferous forests. However, stations have yet to be added to areas with high monitoring potential of grass & shrubland (see Fig. 6c) which is evident also by its continued relative underrepresentation (see Fig 8b).

The flux-view similarly indicates a better representation within the previously unmonitored countries. On European scale, the extension will give a fairer network view if underrepresented fluxes are sensed more than the relative overrepresented fluxes. This is the case with mixed forest fluxes and to a lesser degree with grass & shrubland fluxes, whereas broadleaf forest fluxes are more underrepresented in the extended network. Figures 6a and 6b show how the extended network improves the

monitoring of broadleaf forest especially by the inclusion of the Hungarian station (HUN), but also that great monitoring potential in south-eastern Europe remains. The extension of the German network with Schauinsland (SSL) in the south-western corner of the country mainly improves the representation of mixed and broadleaf forest fluxes (see Fig. 7b).

## 4 Discussion

This study offers an overview of the ICOS network's capacity to monitor the European carbon flux landscape by creating a lens with ICOS station footprints from the summer of 2020. Although the network includes stations in 14 European countries with extensive footprints that transect borders, there are several countries (e.g., Greece, Serbia, and Slovakia) to which the current network is virtually blind. By contrasting the network view with an equal view of Europe and individual European countries, skewed representation of certain land cover types and land cover fluxes are exposed. To plan for network expansion

in relatively well-monitored countries, the analyses should target a specific country, or even a specific region within a large country. As exemplified with Germany, there are still monitoring gaps and unbalanced sensitivity across ecosystems even in member countries with many stations.

    Our results are subject to uncertainties associated with the models we use for footprint calculation (STILT) and the creation of

biogenic flux maps (VPRM), as well as study-specific decisions. FLEXPART (Pisso et al., 2019) is a different footprint model that will be implemented at the Carbon Portal and will allow comparisons for users of our tools that are published along with this study. A study-specific decision is to create network footprints with station footprints that have been limited to the cells with highest influence that add up to 50% of the total sensitivity. This follows the approaches of Henne et al. (2010) and Oney et al. (2015) and is intended to focus our work on the significant influence. In a sensitivity test, we also used the full station

footprints (100%) and found our results robust for Europe, and for countries that have high shares of the network's monitoring capacity. Another choice was to use footprints for the summer (JJA) of the year 2020, with the assumption that they are representative across different years, stations, and networks. However, anomalous meteorology flow or extreme weather causing especially high or low photosynthetic activity could violate that assumption. For the years 2018-2021 we found a similar representation of the different land cover types as in summer 2020. Exceptions include less pronounced

underrepresentation of broadleaf forest fluxes and larger overrepresentation of coniferous forest fluxes on European scale. For Germany, the results are even closer with the most notable difference in the underrepresentation of mixed forests. In the Carbon Portal online tool an analysis with more computationally efficient time-step aggregated footprints is offered and this approach was used for the other years (see Sect. 6). The differences to the approach used in this study are also discussed in more detail there.


The inlet heights chosen for the footprint model is another aspect that might impact the results. In case of multiple inlet heights at a station, the highest level was selected because the top-level measurements have the largest footprints and are generally chosen to provide measurements for regional and global inverse modelling systems. By choosing a lower

inlet height one could potentially focus the view on specific land types. For stations with an air inlet close to the ground the sensitivity can be tenfold that of a mountain station. These large differences in sensitivities between stations are evident in the network maps; the southernmost station Lampedusa with an inlet height of 8 meters (see Table B1) has strong local influences represented by saturated colors close to the station (Fig. 4a; Fig. 9a). Signals at such low-inlet stations are potentially larger and would in turn have greater influence on resulting monitoring potential map. This is an important aspect to keep in mind when new station footprints are created and tested in networks. In our tools, users can make their own decisions in respect to sensitivity threshold, study-period, and inlet height of stations.

In contrast to formal quantitative network design (QND) techniques, our approach does not suggest station locations for an optimal network. However, normally the metric for considering potential station location in QND studies, such as the previously mentioned Nickless et al. (2020), is reduction in uncertainty of underlying carbon fluxes and tend to cluster around the station like footprints. In our approach, the footprints are not analysed in terms how uncertain the underlying fluxes are, but the derived monitoring potential maps will highlight areas where the highest under-monitored flux activity is. This normally coincides with where the prescribed uncertainties are highest. Furthermore, the simplicity of our approach makes it possible to present users with tools to consider any network configuration within Europe. The tools will be open for future development, and it will be possible to introduce new and updated data layers to overlay with the footprints.

## 5 Conclusion

We evaluated the ICOS network to gain insight of what land cover types and land cover associated fluxes it represents to reveal strengths, weaknesses, and potential gaps. The network is formed by the combined views of the individual ICOS atmospheric stations and has a high monitoring capacity in central and northern Europe where stations are relatively concentrated. The stations pick up signals throughout the heterogeneous European flux landscape and show a large variation of sensitivities, with a larger sensing capacity for biogenic fluxes than for anthropogenic emissions during the study period of summer year 2020. The summer is most interesting from a biosphere monitoring perspective and 2020 has also proven representative for longer time-periods in terms of our conclusions about the network. As a network, the land cover view relatively overrepresents coniferous forests and cropland as opposed to broadleaf forests and grass & shrubland in the European context. Country-scale considerations reveal a more uneven representation in some countries, such as Norway where coniferous forests are mainly within the network view. The land cover view is not necessarily the same as the GEE-view due to $CO_2$-fluxes being spatially and temporally heterogeneous within European land cover types and further indicates that highly active broadleaf forest fluxes are missed. Relative monitoring potential maps indicate a high potential for this flux, and similarly underrepresented grass &

shrubland flux, in south-eastern Europe. The presented views of the carbon flux landscape from the ICOS network will change as new countries are joining the network, national networks are changing, and the underlying flux landscape is changing. With

careful planning of new station locations, we can hopefully make the most of measurements in the ongoing task of improving our understanding of the carbon flux landscape.

## 6 Code availability

A folder with scripts and notebooks to re-produce the analyses of the study also for other stations, countries, and networks has been published (Storm et al., 2022). There are dependencies with files on the ICOS Carbon Portal Jupyter Service

(https://www.icos-cp.eu/data-services/tools/jupyter-notebook) and the notebooks can only be run there. Hypothetical or actual stations with existing footprints can be used directly, and footprints for additional locations can be produced using the Carbon Portal's STILT on demand calculator (https://stilt.icos-cp.eu/worker/).

## 7 Data availability

Datasets used as input for the analyses are listed throughout the paper. Station footprints used to create network footprints are

published with persistent identifiers and available for download at the Carbon Portal (https://data.icos-cp.eu/). The figures and tables in the result section have been created with the published scripts.

*Author contributions.* IS, UK, AV, and WP designed the study. IS developed the Jupyter notebook package together with CD and UK. IS, WP, and UK prepared the paper, with contributions from all co-authors.


*Competing interests.* The authors declare that they have no conflict of interest.

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
