# Peer review of "A view of the European carbon flux landscape through the lens of the ICOS atmospheric observation network"

_Atmospheric Chemistry and Physics, 2022_

## Author Comment (AC1)

**RC1: 'Comment on acp-2022-756', Anonymous Referee #1, 15 Jan 2023**

The manuscript 'A view of the European carbon flux landscape through the lens of the ICOS atmospheric observation network' by Storm and co-authors discusses and characterises the ability of the ICOS atmospheric monitoring network to representatively observe the European carbon budget. The study is based on atmospheric transport model simulations, land-cover maps and pre-calculated biospheric fluxes. The paper introduces a few new metrics that can be used for network design/characterisation and help with the identification of underrepresented land cover types and regions. The study is mostly based on established methods (transport simulations, flux calculations, emission inventories) and while some of the conclusions as to where the network is currently lacking additional sites seem obvious, the authors offer a more quantitative tool to support such assumptions.

Language, structure and technical quality of the manuscript are mostly adequate for publication. There are a few aspects where the clarity of the manuscript could be improved and at least two major points which should be addressed to strengthen the conclusions of the paper. Once these and further minor comments are addressed the manuscript should be fit for publication.

We thank the reviewer for their assessment of our manuscript. The individual points made are addressed below. Please note that the line numbers in the reviewer comments refer to the first version of the paper and our responses to the latest version of the paper.

**Major comments**

Comparison to real world data:

The study is solely based on bottom-up anthropogenic, oceanic and biogenic $CO_2$ fluxes combined with atmospheric transport simulations. Table 1 presents mean model simulated contributions to $CO_2$ mole fractions at the different sites of the ICOS network, but there is no validation of how well these mean mole fractions compare to observed mole fractions. Even more interesting would be an indication of how well the simulations fit the individual afternoon observations. Demonstrating the general ability of the model to capture the observed mole fractions for one or a few sites would strengthen the credibility in the model system and all following analysis and conclusions on the network. If such comparisons already exist for the presented model system, it would be sufficient to refer to these results. Otherwise, I would suggest to include a comparison for at least the site that is already discussed as an example, Hyltemossa.

We agree with the reviewer that this type of comparison would be useful and strengthen the credibility of the modelling system. We have considered the agreement between modelled and measured concentrations for all stations in the "current" ICOS network, and the results are shown in the table below. For year 2020, the correlation coefficient (r) is on average 0.82. The reviewer expressed interest in the afternoon hours, and if only these (12:00 and 15:00 UTC) are considered, the average correlation coefficient is rather 0.86. We choose to mention the correlation given all hours of the day we have model results for (0:00, 3:00, 6:00, 9:00, 12:00, 15:00, 18:00, 21:00) as we do not exclude any hours in our study. However, we include both selections in Appendix B:

| STILT station | Sampling height | r (all hours) | r (12, 15 UCT) | std (all hours) | Std (12, 15 UCT) | RMSE (all hours) | RMSE (12, 15 UCT) |
|---|---|---|---|---|---|---|---|
| BIR075 | 75 | 0,88 | 0,92 | 3,41 | 3,21 | 4,32 | 3,69 |
| CMN760 | 8 | 0,88 | 0,90 | 3,45 | 3,54 | 3,53 | 3,54 |

| | | | | | | | |
|---|---|---|---|---|---|---|---|
| GAT344 | 341 | 0,88 | 0,89 | 4,31 | 4,22 | 4,56 | 4,43 |
| HEL110 | 110 | 0,91 | 0,92 | 5,47 | 5,28 | 7,01 | 6,77 |
| HPB131 | 131 | 0,77 | 0,79 | 5,90 | 5,85 | 6,24 | 6,01 |
| HTM150 | 150 | 0,90 | 0,93 | 3,83 | 3,32 | 4,36 | 3,49 |
| IPR100 | 100 | 0,70 | 0,85 | 11,39 | 8,08 | 15,46 | 8,97 |
| JFJ | 5 | 0,81 | 0,81 | 3,92 | 4,06 | 4,10 | 4,24 |
| JUE120 | 120 | 0,57 | 0,72 | 12,74 | 11,04 | 12,77 | 11,62 |
| KIT200 | 200 | 0,68 | 0,82 | 11,18 | 9,01 | 13,22 | 9,97 |
| KRE125 | 125 | 0,79 | 0,86 | 5,74 | 4,72 | 6,60 | 4,94 |
| LIN099 | 98 | 0,71 | 0,86 | 8,21 | 5,12 | 8,82 | 5,33 |
| LMP | 8 | 0,92 | 0,92 | 2,11 | 2,12 | 2,25 | 2,21 |
| NOR100 | 100 | 0,84 | 0,92 | 4,69 | 3,51 | 5,65 | 3,73 |
| OPE120 | 120 | 0,82 | 0,87 | 4,58 | 3,79 | 4,97 | 3,89 |
| OXK163 | 163 | 0,77 | 0,75 | 5,38 | 6,04 | 5,49 | 6,12 |
| PAL | 12 | 0,89 | 0,95 | 3,46 | 2,73 | 3,53 | 2,82 |
| PRS | 10 | 0,84 | 0,87 | 3,35 | 2,78 | 3,41 | 2,79 |
| PUI084 | 84 | 0,89 | 0,95 | 4,04 | 3,14 | 4,64 | 3,26 |
| PUY | 10 | 0,83 | 0,86 | 3,33 | 3,05 | 3,44 | 3,06 |
| SAC100 | 100 | 0,64 | 0,83 | 9,77 | 6,27 | 9,90 | 6,26 |
| SMR125 | 125 | 0,92 | 0,95 | 3,49 | 2,92 | 4,13 | 3,14 |
| STE252 | 252 | 0,84 | 0,88 | 5,67 | 5,11 | 6,34 | 5,53 |
| SVB150 | 150 | 0,92 | 0,95 | 3,16 | 2,79 | 3,77 | 3,11 |
| TOH147 | 147 | 0,83 | 0,83 | 4,72 | 4,82 | 4,72 | 4,82 |
| TRN180 | 180 | 0,79 | 0,86 | 4,61 | 3,73 | 5,01 | 3,83 |
| UTO | 57 | 0,91 | 0,90 | 3,32 | 3,43 | 3,71 | 3,85 |
| WAO | 10 | 0,70 | 0,60 | 7,72 | 10,15 | 8,30 | 10,66 |
| ZSF | 3 | 0,87 | 0,88 | 3,54 | 3,48 | 3,65 | 3,55 |

**Table 1. Correlation between modelled (STILT Carbon Portal implementation) and measured CO₂ concentrations at the stations in the current ICOS network year 2020. All hours with model results are considered; 0:00, 3:00, 6:00, 9:00, 12:00, 15:00, 18:00, 21:00.**

[Figure]

**Figure 1.  Timeseries for Hyltemossa (150 m) of modeled CO2 concentrations from the STILT Carbon Portal implementation and measured CO₂ concentrations year 2020. All hours with model results are considered; 0:00, 3:00, 6:00, 9:00, 12:00, 15:00, 18:00, 21:00.**

This information has been added to Sect. 2.2 of the paper:

"The total modelled concentrations can in turn be compared to measured concentrations to assess the performance of the modelling system; With an average correlation coefficient of 0.82 for all stations in the current network year 2020 there is generally a good agreement. Comparisons for the individual stations can also be found in Appendix B."

Respiration fluxes:

A major part of the discussion is focused on the network characterisation with respect to summertime gross primary production (or here gross ecosystem exchange, GEE).

However, ignoring respiration for the network characterisation seems like abandoning the study half-way. Respiration fluxes show a distinctly different temporal evolution as compared to GEE fluxes. Relative contributions from different land cover types may differ between respiration and GEE. Average footprints differ between summer and winter (as shown in Fig. 2). All these points indicate that conclusions drawn in terms of representatively monitoring GEE do not necessarily apply to respiration or NEE. This is already demonstrated by the flux contributions at the site Hyltemossa (Fig. 3), where contributions by land cover differ strongly between GEE and respiration as well as between summer and winter. If the aim of the ICOS atmospheric network is the observation of the complete carbon budget and not just the summertime $CO_2$ uptake, the present analysis should integrate a discussion of respiration and/or NEE.

We recognize the reviewers concern with our focus on GEE and specifically the summer. Tied to other comments by the reviewer, we have made this clearer and state that the conclusions about the network are with regards to this. During the study, we considered the inclusion of respiration but decided to proceed with only GEE and provide our rationale for this below. However, we will make it possible to choose between respiration and GEE in our online tool in its next release.

1. In terms of NEE, this is what we state in the paper (Sect. 2.5).

"We use GEE (Gross Ecosystem Exchange) rather than NEE (Net Ecosystem Exchange = GEE + Respiration) as footprint-weight to prevent nearly cancelling photosynthesis and respiration signals to influence the network view. The GEE-view thus highlights areas with high biogenic activity which are especially important to monitor; high activity generally means greater uncertainties in the current estimates and higher potential for long-term carbon storage." So recognizing that NEE is small and changes sign, we are left with the choice of considering GEE or Respiration as proxy for ecosystem sensitivity of the network.

2. Our focus on GEE, and not respiration.

The large signals associated with different land cover types that have high GEE values (large sink) in general also have high respiration values (large source). This is expected, as respiration is a function of carbon in soils and litter, which is large for highly active vegetation (high GEE). Correlation coefficients for both, associated with different land cover types, range from 0.74 for pasture to 0.96 for coniferous forest. Note that the different diurnal cycle in respiration compared to GEE cancels out (see our signals in Table 1) in the average over the whole time-period (summer 2020).

The correlation between high GEE and respiration is also evident spatially when we evaluate how different our relative monitoring potential maps would look like if respiration would be used instead of GEE: For Europe, all the land cover classes, the correlation coefficient is over 0.8. A general pattern in the maps is that the monitoring potential is slightly more spread out when it comes to

respiration. This is because of the relative nature of the monitoring potential maps, in which the most active locations (in GEE especially forests) get relatively more weight. This is as intended, as highly active vegetation should receive extra attention because of their influence on the final carbon balance.

| | Broad leaf forest | Coniferous forest | Mixed forest | Other | Grass and shrubland | Cropland | Pastures | Urban |
|---|---|---|---|---|---|---|---|---|
| GEE summer 2020 | -16,5 | 0,9 | -0,4 | 40,7 | -53,7 | 13,6 | 7,7 | 63,9 |
| RESP summer 2020 | -8,9 | 5,3 | -8,4 | 50,2 | -53,6 | 7,3 | -2,4 | 62,0 |

**Table 2. The over- (+) or under (-) representation of the current ICOS network within Europe (see paper Sect. 2.5). Results given for respiration and GEE summer year 2020.**

3. Our focus on summer, rather than the whole year

Our rationale for the focus on summer is that this is the time of the year with highest biogenic activity and when it is most important that we are able to monitor the ecosystems. Moreover, this is the gross flux that dominates during daytime when the data is used for inverse modelling. However, we see the value in more carefully consider the sensitivity of the results to the use of a different time-period.

| | Broad leaf forest | Coniferous forest | Mixed forest | Other | Grass and shrubland | Cropland | Pastures | Urban |
|---|---|---|---|---|---|---|---|---|
| GEE summer 2018 | -5,7 | 2,8 | -3,9 | 70,3 | -54,1 | 4,7 | 12,6 | 72,8 |
| GEE summer 2019 | -1,7 | 5,1 | 0,8 | 70,8 | -50,5 | 12,2 | 9,6 | 71,2 |
| GEE summer 2020 | -12,6 | 8,9 | -3,7 | 56,8 | -52,9 | 8,6 | 7,2 | 59,5 |
| GEE summer 2020 backward* | -16,5 | 0,9 | -0,4 | 40,7 | -53,7 | 13,6 | 7,7 | 63,9 |
| GEE summer 2021 | -9,0 | 12,2 | -2,6 | 63,2 | -51,4 | 13,4 | 12,8 | 65,4 |

**Table 3. The over- (+) or under (-) representation of the current ICOS network within Europe (see paper Sect. 2.5) for summers years 2018-2021. *Updated approach which is computationally heavier, and we only have results for year 2020 currently. See replies to RC2 for more details**.

The choice to specifically use summer 2020 is discussed with the results for summers years 2018-2021 in mind):

"For the years 2018-2021 we found a similar breakdown of representation as in summer 2020. Exceptions include less pronounced underrepresentations of broadleaf forest fluxes and larger over-representations of coniferous forest fluxes on European scale. For Germany, the results are even closer with the most notable difference in the underrepresentation of mixed forests. In the Carbon Portal online tool an analysis with more computationally efficient time-step aggregated footprints is offered and this approach was used for the other years (see Sect. 6). The differences to the approach used in this study are also discussed in more detail there. "

**Minor comments**

L12, L75: When introducing the footprints for the first time, it may be helpful to call them 'concentration footprints' to distinguish them from 'flux footprints' as used in the eddy covariance community.

The suggestion was implemented.

L13: 'European flux landscape'. Not very specific. Better 'carbon flux' or even 'CO2 flux'.

"European flux landscape" changed to European $CO_2$ flux landscape.

L16: Same as previous: 'anthropogenic CO2 emissions'. In general, a note in the conclusions could be added that the results obtained here are exclusively valid for the CO2 flux landscape. Other parameters observed by the network (CH4, N2O) exhibit a completely different flux distribution and a similar analysis could be carried out in order to characterise their gaps and monitoring potentials.

The suggestion was implemented.

We decided to stress the validity of the study for the $CO_2$ flux landscape already in the introduction:

"We chose in this study to quantify and summarize the capacity of the ICOS atmospheric observing network to sense the underlying $CO_2$ flux landscape. Other parameters observed by the network including $CH_4$, $N_2O$ exhibit different flux distributions and would therefore need separate analyses to characterise their gaps and monitoring potentials. "

Fig 1 (and others): Although the spatial reference system is noted in the legend, their are no axis labels which would allow for geographic referencing. Please add.

The suggestion was implemented for all maps.

In caption, please mention source of land cover classes given in the figure.

The reference to the dataset was added.

L57: Be more specific. What kind of measurements?

The text has been updated and reads:

"Measurements of atmospheric trace gas concentrations have traditionally been collected at remote islands, mountain tops, or other locations at large distance from direct emissions or uptake, to find well-mixed conditions that represent background atmospheric levels (Conway et al., 1994)."

L66: European climate zones stretch form sub-tropical (dry summers) through temperate (Central Europe) to sub-arctic. Please rephrase accordingly.

The text is in accordance with the map in our reference (Beck et al., 2018). However, we realize that "temperate with dry and hot summers" sound a bit unfamiliar and now use "Mediterranean with dry and hot summers" instead:

Europe has multiple climate zones ranging from Mediterranean with dry and hot summers in the South through temperate to cold Northern sub-Arctic climate without a dry season according to the Köppen-Geiger classification (Beck et al., 2018).

L69: Replace 'has more' by 'dominated by'.

The suggestion was implemented.

L95: There is no absolute measure of sensing capacity, I would argue that the metrics introduced here should be called 'more quantitative' instead of 'quantitative'. Or at least add that the introduced metric remains semi-objective as you could have used a slightly different metric instead.

We agree and have implemented the suggestion to call it "more quantitative" instead of quantitative.

L107f: Please refer to the 'Code availability' section here. I tried to follow the links and get the jupyter notebook to run. However, I got a 'page not found' error when trying the link for Storm 2022 in the list of references. Hence, I was not able to check whether there are working tools behind this or not. In the ideal case, getting your tools to run should not cost a whole day to set it up.

Unfortunately, the link in the original PDF also included the page number of the paper, which meant clicking it did not work. The preprint and new version of the paper have working links. We are sorry about this.

Furthermore, an up to date version of the tool is also available on the Carbon Portal JupyterHub: login with email address and current password "francis" (the update may change, see instructions here: https://www.icos-cp.eu/data-services/tools/jupyter-notebook/exploredata-password). If the JupyterHub is used, there is no setup of the tool required and users can start their runs immediately.

Sect. 2.1: In this section it does not become clear for which period the analysis was done. Later it is mentioned that the analysis was done for summer (winter?) 2020. How representative is a single summer? Was the summer 2020 rather typical are characterised by extremes?

We have added the following text to the end of section 2.1:

"Footprints for summer (JJA) were used for the subsequent analyses. Additionally, footprints for winter (DJF) year 2020, were used only in the analysis of individual station for (Sect. 3.1) exemplified with Hyltemossa (Sect. 3.1)"

In terms of the representativeness of a single summer, please see the response to the respiration discussion. There are naturally differences between summers, and we are more carefully stressing that the conclusions are specifically for summer 2020 in the updated version of the paper.

L122: What was the spatial resolution of ECMWF data used?

We have added this information to the paper:

"Meteorological conditions drive the transport and are represented by three-hourly operational ECMWF-IFS analysis/forecasts at 0.25 degrees resolution."

Sect 2.2: Maybe change title so that it includes the description of the utilised fluxes, for example "CO2 fluxes and simulated signals at the stations"

The suggestion was implemented.

L130: This is a rather old version of EDGAR. Why is it used instead of the more previous releases?

We use the Carbon Portal Footprint tool which is indeed based on an older version of EDGAR. However, the emissions have been extrapolated to 2020 using energy statistics and the dataset has also been temporally disaggregated to hourly resolution. For more detailed information, readers can consult the description of the set-up which is included as a reference in the paper:

Karstens, U (2022): https://meta.icos-cp.eu/objects/XX3nZE3l0ODO9QA-T9gqI0GU.

L135ff: Not quite clear how this was done? Was VPRM run twice (once with SYNMAP and once with HILDA land-cover) or simply on SYNMAP and the remaining land-cover analysis was done with HILDA only? How does resolution of these two land cover datasets relate to the output resolution of STILT? Where is SYNMAP data published? In a quick search I could only find the paper but no link to the data (as claimed in the data availability section).

We realize how this needs to be better explained and have re-worked the text (see below). In short, the VPRM is only run once, and we use the output published at the Carbon Portal (https://doi.org/10.18160/VX78-HVA1). The creators of the VPRM biosphere fluxes refer, like we do, to the paper by Jung et al., 2006:

Gerbig, C.: Parameters for the Vegetation Photosynthesis and Respiration Model VPRM (Version 1.1), ICOS-ERIC – Carbon Portal [data set], https://doi.org/10.18160/R9X0-BW7T, 2021a.

SYNMAP has been accessed upon a request to the authors (M. Jung). The Vegetation fraction maps for the STILT resolution (1/8 x 1/12 degrees) were created based on aggregated groups of the 1km resolution original SYNAMP land cover. To again disaggregate the land cover, we alternatively re-create the timeseries using HILDA because we see advantages with the dataset compared to using the original SYNMAP (see details below). The resolution of the HILDA land cover is also 1km, and in the same way as the creators of the VPRM fluxes, we prepared alternative vegetation fraction maps for the combination with STILT footprints.

"The biosphere-flux derived signals computed during the footprint calculation are attributed to groups of aggregated SYNMAP (Jung et al., 2006) land cover categories, which is also the map used to parameterize the VPRM model. The aggregation used within the STILT Footprint Tool makes for broad categories such as "crop and tree", which is attributed to over half of the land area in the model domain. To again disaggregate the land cover, we alternatively re-create the timeseries of biogenic signals by combining the footprints for hourly backward time-steps with the temporally resolved flux maps but attribute the resulting biogenic signals to the different land cover categories in HILDA (Winkler et al., 2020). The HILDA land cover is a synthesis product built on multiple heterogenous datasets including several satellite data products which were published only after SYNMAP was created (Winkler et al., 2021). Another advantage is that HILDA map used in this study represents year 2018 as opposed to year 2000. "

L145: It could still be interesting if by choosing a lower inlet height one could focus the view on specific land types. However, I agree that by selecting the more convective afternoon situations, differences my not be too large as compared to the uppermost inlet height. Maybe something for future analysis.

Indeed, to focus on specific land cover types is something that could be the reason for choosing a different inlet height. We have added a note about this in the discussion:

"In case of multiple inlet heights at a station, the highest level was selected because the top-level measurements have the largest influence areas (footprints) and are generally chosen to provide measurements for regional and global inverse modelling systems. By choosing a lower inlet height one could potentially focus the view on specific land types."

For the interest of the reviewer, we can share that we had a look at what the difference would be for Hyltemossa's lower inlet heights (70 m and 30 m) compared to the highest inlet (150 m) and found increasing shares of coniferous forests in the average footprints with decreasing inlet heights. The

generally more local sensing associated with lower inlet heights agree well with this picture as coniferous forests is found in abundance in the area surrounding Hyltemossa.

L155: The 50 % are arbitrary. Why not use the complete footprint here to calculate the contributions from different land cover? How large would the difference be?

In the paper we provide the following rationale for using the 50% footprints:

"The hourly backward time-steps footprints for the individual stations in the network are reduced to grid cells with the highest sensitivity values that in combination add up to 50% of the sum of the hourly backward time-step footprints sensitivities. This follows the approach of Henne et al. (2010) and Oney et al. (2015) and is intended to emphasize areas with significant local influence."

Furthermore, the uncertainties in the sensitivity values derived from the hypothetically dispersed particles increase further backward in time, when they are further from the station. This is reflected in a dynamic aggregation of cells to final footprints with large areas far from the station attributed with the same sensitivity values. For our study, this means generally small differences in using the 50% footprints compared to the full footprints as much of the sensitivity is spread out evenly over areas of diverse land cover. A sensitivity test was done with timestep-aggregated footprints and shows the following differences in representation:

| Component | Broad leaf forest | Coniferous forest | Mixed forest | Other | Grass and shrubland | Cropland | Pastures | Urban |
|---|---|---|---|---|---|---|---|---|
| GEE summer 2020 | -16,5 | 0,9 | -0,4 | 40,7 | -53,7 | 13,6 | 7,7 | 63,9 |
| GEE summer 2020 100% footprints | -14,30 | 3,99 | -7,72 | 38,60 | -43,88 | 5,18 | -1,11 | 49,79 |

**Table 4. The over- (+) or under (-) representation of the current ICOS network within Europe (see paper Sect. 2.5). Results given for GEE summer year 2020 with 50% footprints and 100% footprints.**

L159: Why take the maximum instead of the sum? In the real world multiple sites would receive information from the grid cell, which adds to the information available to any inverse modelling approach. By using the maximum here, a general underestimation of the total network sensitivity results.

When footprints of stations overlap, we take the maximum sensitivity to the cell of the two stations. Rather than when taking the sum of the overlap, it focuses the analysis on representation of fluxes spatially. With our choice, placing a new station close to an old would not double the capacity to monitor fluxes. We agree that in a quantitative estimate of $CO_2$ fluxes using such footprints, one would indeed use the sum of sensitivities and derive a flux with smaller uncertainty under an overlapping footprint.

However, we found that when considering the sum instead of the maximum for our study, the differences are small because there is anyway limited overlap between the atmospheric stations in our network, as they are generally at quite a large distance from each other. We for example looked at how much new stations caused overlap with the "current" network:

"Overlap (see Sect. 2.4) with the sensing capacity of the current network is close to zero for Spanish and Irish stations with respect to their 50% summertime footprints, and well-below 10% for all other

added stations with exceptions for German Schauinsland (SSL) and Polish mountain-station Sněžka (SNZ)."

L163: What is quantified here? Additional area covered? Differences in land cover contributions? Please clarify.

This is related to the last point and has been clarified in the text. What is quantified is the "loss" in network sensitivities from taking the maximum rather than the sum (last reviewer point):

"To estimate the overlap in sensitivity between a current network footprint and the 50% footprint of a station that is included in the extended network, the effect of its inclusion in an updated network footprint is analysed; the difference between the spatial sums of the two network footprint sensitivities is compared to the spatial sum of the 50% station footprint. If there is no overlap between the current network and the footprint of the station joining, the difference between the two network footprints is the same as the sum of the (50%) station footprint."

L168: Same question as above: Are VPRM fluxes based on HILDA?

No, VPRM fluxes are not based on HILDA. This is addressed in the reply to the question above (L135ff).

L169: The abbreviations 'GEE' and 'NEE' are never properly introduced.

We have added proper introductions to the abbreviations where they first appear.

L173: Also see major comment above: It would still be interesting to see if results would be similar for a respiration view which would need to include a whole year of simulations. As you say respiration is similarly important for the carbon budget and the network hence should equally be representative for respiration. Even for GEE it would seem beneficial to include a whole growing season and not just the summer as uptake will differ in timing between different land-cover, especially crops versus forest, and climate region, early growing season in the South limited by water availability in the summer versus northern climate zones with late onset of growing season. Only focusing on summer may introduce a bias here as well.

This was addressed in the reply to the major comment.

We encourage users to further test different time periods with our online tool.

L185ff: These equations need to be properly typeset and variables a to h defined as such. It is not clear over which dimensions the individual variables run (space, time, land cover) and over which of these the sums are defined.

We agree that this is needed to make it more clear and have updated the equations. They are provided at the end of this document.

L189: To me it is not really clear at this point why you need the footprints here. Is the area mean/total not simply defined by averaging/summing the proxy data over the area? Or do you additionally limit to the 50% footprint? Which I think would not make sense since you would limit the equal view already to what is within the direct view of the network. Or is the total simulated sensitivity redistributed equally to the target area?

The reviewer is referring to the section about the creation of the equal view footprint (Eq. 2 at the bottom of the document). With the updated text and the properly typeset variables we hope this is now clear. We try to also give the reviewer an answer here:

We combine "equal view network footprints", based on the original network footprints, with underlying data to arrive at what signals the network would pick up if the monitoring potential was spread out evenly. This feels intuitive to us as we can change what the network represents by consciously adding stations in areas with high monitoring potential of desired ecosystem types. However, this indeed gives the same results as using average proxy data in combination with the network footprints.

L195: Why is this called a mask? Isn't it simply the fraction in each grid cell. Potentially reaching from 0 to 100 %

It is indeed the fraction in each cell and the text has been updated in accordance.

L196: So e and f are obtained for each land cover type, correct? Please indicate this by a running index on those variables that are defined for different land cover. Supposedly c and d as well.

Yes, indeed. The equations are now properly typeset, and it is hopefully clear (provided at the bottom of this document)..

L207: Got me all confused here. f > e would give negative h according to equation 4.

The absolute value of GEE was used. Hopefully it is all clear now with the updated equations (provided at the bottom of this document).

Table 1: Some columns are given with 1 others with 2 significant digits. For example the 'Residential' contribution is 0.0, 0.1, or 0.2 everywhere. Difficult to make out differences from this. Should GEE not be given with a negative sign to indicate uptake? I don't think this is ever clearly spelled out anywhere.

We have updated the table which now have two decimals and GEE is indicated as negative.

L222f, L233f: That conclusion is too general since only GEE is looked at. The picture would, most likely, look very different if NEE would be evaluated. Please consider that the CO2-only observations cannot distinguish between anthropogenic, uptake and respiration. Hence, only concluding from GEE that the network is mostly sensitive to biogenic is not valid. The conclusion may still be correct for summer (but even that is not mentioned here), but you would need to show with a NEE view!

Please see our response to the second major comment.

L255: These cites are not indicated in the figure. Please add for reference.

The reviewer is referring to Malmö and Copenhagen which have now been added to Fig. 2a.

L261ff: Is this analysis based on land cover alone (not fluxes)? Is the land cover weighted by footprint sensitivity before calculating the shares or is just the area analysed? Would be nice to add two bars for the total to Fig 2b (the analysis by direction) as the text seems to be discussing the total rather than any specific direction.

Yes, it is based on land cover alone and weighted by footprint sensitivity.

We have updated the legend text to make this clearer:

"summer (JJA) and winter (JFD) land cover shares weighed by the seasonal footprints split by direction."

The suggestion to add two bars for the summer and winter total has been implemented and updated graph looks like this:

[Figure]

The reviewer is referring to this paragraph:

"During the winter, biosphere respiration of CO2 is almost as large signal as the anthropogenic contribution, which is dominated by energy production, followed by residential. Interestingly, emission sources within Sweden only contribute about 23% of the anthropogenic signal at the station, and the previously mentioned Oresund region slightly more. The remainder is transported from emission sources further away, emphasizing the importance of long-range transport."

Yes, this is true for Hyltemossa during winter 2020. This is a station located quite far north (lat: 56.1, lon: 13.4) and there is little biogenic activity during this time of the year here. More interesting from a monitoring perspective when it comes to the biosphere is what is going on when the biosphere is active. We have made sure to clearly state that our analysis focus on the summer, when the biosphere signals are generally much larger than the anthropogenic signals. We also point to that sub-sampling of timeseries as a way to avoid measurements that are contaminated in subsequent modelling:

"However, it is important to remember that the signal averages include peaks in anthropogenic signals during particular hours especially when the wind transports air from large point source emitters. For example, German station Jüelich (JUE) is located only 10 km from a coal-fired power plant that accounts for about 4% of Germany's total emissions (E-PRTR) and whereas the average signal is 5.6 ppm, it is below 1.0 ppm about 20% of the time. Careful sub-sampling of time series, as suggested also by Oney et al. (2015), could allow for either avoiding anthropogenic influence or concentrating on its analysis."

Fig. 4: All the labels are way to small. Also the shading for the footprints is not necessarily well chosen and does not offer many details on the actual distribution as it mainly shows the location of

the sites. It would also be interesting to discuss some of the apparent differences between sites. Some stick out with very large sensitivity in the surroundings, others are hardly visible.

We have increased the font of all figures.

We have discussed the way the footprints look and concluded that we will keep the current shading. An alternative would be to use a logarithmic scale which would lessen the emphasis on the areas around the station locations. However, we want to be careful with this not to overstate the network monitoring capacity.

In terms of the differences between stations, we have added the following:

"For stations with an air inlet close to the ground the sensitivity can be tenfold that of a mountain station. These large differences in sensitivities between stations are evident in the network maps; the southernmost station Lampedusa with an inlet height of 8 meters (see Table A1) has strong local influences with saturated colors close to the station (Fig. 4a; Fig. 9a). Signals at such low-inlet stations are potentially larger and would in turn have greater influence on resulting monitoring potential maps."

L287f, Fig. 4b: I don't see the big difference for coniferous forests. I see differences for grass/shrub and also for urban, but the forests seem to be represented rather well. You could use percentages in the text to underline your point.

The difference in share of coniferous forests in Europe compared to the network is indeed not as large for coniferous forests as for the land cover classes grass & shrubland and urban. This is what figure 4b looks like (and it is Europe, furthest to the right, is discussed):

[Figure]

We decided against using percentages in the text because it gives the impression of a quantitative analysis: it is qualitative analysis and the result will be dependent on the choices we have made.

The network share of coniferous forests is 20.6% compared to 17.8% within Europe. The difference for grass & shrubland is indeed larger with a network share of 4.5% compared to 10.3% within Europe.

The difference for the land cover type "urban" is larger than that for coniferous forest (9.7% network share compared to 5.9% within Europe), but we choose to focus on the non-urban land cover classes here. In the flux discussion we mention the over-representation of the urban land cover:

"The overrepresentation of sensing of fluxes associated with the land cover type urban (see Fig. 5b), despite the ICOS network targeting natural fluxes, is explained by the relatively high density of stations in central Europe; countries with highest sensitivity per area unit (Switzerland, the Netherlands, and Germany; see Fig. 4b) are also counties with some of the highest population densities."

Fig. 5: In b, do the upper and lower bars refer to current and extended network, respectively? Please add information to caption or plot.

The caption for the plot(s) – also for Germany (Fig. 8) – were updated:

"Figure 5: (a) Share of flux (GEE) per land cover within Europe compared to the network GEE-view for the current and extended networks within Europe. (b) The over- (+) or under (-) representation of the current ICOS network (upper bars) compared to the extended (lower bars) ICOS network within Europe (see Sect. 2.5)."

L304f: Where do we see this? Is this in Fig 5?

We have updated the text to reflect where this can be seen:

"The underrepresentation of broadleaf forest fluxes (see Fig. 5b), despite a fair LC-view (see Fig. 4b), means that broadleaf forests outside the focus of the network are relatively more active than those currently sensed"

L312f: Could you please add references to the figures where the individual points can be seen? 'underrepresented flux' in Fig 5? 'Monitoring potential for Serbia and Croatia' in Fig 6c,d?

We have updated the text to reflect where this can be seen:

"Grass & shrubland is the most underrepresented flux (see Fig. 5b) and shows greatest potential for monitoring in Serbia and Croatia (see Fig. 6b). Within the ICOS membership countries, Scandinavia has the highest potential (see Fig. 6b)."

Fig. 6: Are the same color scales used for the main plot and the German inset? Were these monitoring potentials for Europe and Germany calculated separately. Why would they look so different in the main plot and the inset? What are the actual values (not indicated on the color scale)? How should these be interpreted? I think it would be easier to understand if you don't present the European and German case in one plot, but have two separate plots.

We have updated the maps so Germany and Europe are no longer in the same figure because they indeed have their relative monitoring potentials calculated separately. We see how it is confusing especially as Germany is white in the map showing Europe and has colors in the insert map showing Germany. To clarify how this can be, we have added a short explanation in the separate figure showing the relative monitoring potential of Germany (Fig. 7):

"Same as Fig. 6, but relative monitoring potential within Germany as opposed to Europe. In the European context, Germany is relatively well-monitored and appears to have no relative monitoring potential (white)."

We choose not to use actual values because the "equal view" and the "network view" of the fluxes used to establish the GEE-view is based on network footprints. The resulting values from combining the sensitivities of multiple stations (network footprints) with fluxes only have meaning in the relative sense. How the "equal view" and "network view" is used to create monitoring potential maps can be found in the updated methods section.

L322: Unclear. Do you mean that because of large local fluxes there is little additional signal for regions farther away? Even with the next sentence this is somewhat unclear.

The reviewer is referring to the following text (underlined):

"The overrepresentation of sensing of fluxes associated with the land cover type urban, despite the ICOS network targeting natural fluxes, is explained by the relatively high density of stations in central Europe; countries with highest sensitivity per area unit (Switzerland, the Netherlands, and Germany) are also counties with some of the highest population densities. The relatively high sensing per unit area in central Europe also means that these countries show little or no monitoring potential on European scale. This should not be interpreted as if their monitoring is "complete"; it only means they are well-monitored relative to other areas in Europe. For expansion of national networks, the same approach can be employed on country scale to analyse flux representation and highlight relative monitoring potential which we will illustrate for Germany."

The confusion regarding the concept of monitoring potential is evident also from the last point the reviewer made and we are thankful to have the chance to clarify this. The changes in response to the last point, and the updated equation with proper typeset variable (provided at the end of this document) will hopefully help, and we have updated the text the review is referring to:

"The relatively high sensitivity of the network within these countries compared to the rest of Europe also means that their fluxes are relatively well-monitored and appear low in monitoring potential when entire Europe is considered (see Eq. 2-Eq. 5). This should not be interpreted as if their monitoring is "complete" and for expansions of national networks it is advisable to consider the relative sensing within the individual countries which we will illustrate for Germany."

L336: There is no table 3 in the manuscript. Only the single Table 1! Did you mean that? But how do I see relative contributions there?

We indeed refer to Table 1 and not the non-existing Table 3. The text has been updated in accordance.

Figure 8: Do the two different color shadings represent the same value range? Please clarify by either adding a second color palette or clarify in the caption.

We see how this is confusing and have clarified the different shadings in the caption:

Figure 9: (a) Summer (JJA) 2020 extended network footprint overlaid on the current network footprint. The same level of color saturation of green and blue have the same meaning.

We chose not to add a second color palette as the focus is on the extended network which has the shown palette. The shading for the current network is the same as in the figure where it is shown separately (Fig. 4a).

L352: What does 'overlap' mean here? Is this with respect to the cropped 50 % footprints?

The reviewer is referring to this section which we have underlined with the updates to make the meaning of "overlap" clearer:

"Overlap (see Sect. 2.4) with the sensing capacity of the current network is close to zero for Spanish and Irish stations with respect to their 50% summertime footprints, and well-below 10% for all other added stations with exceptions for German Schauinsland (SSL) and Polish mountain-station Sněžka (SNZ)."

The updated description in section 2.4:

"To estimate the overlap in sensitivity between a current network footprint and the 50% footprint of a station that is included in the extended network, the effect of its inclusion in an updated network footprint is analysed; the difference between the spatial sums of the two network footprint sensitivities is compared to the spatial sum of the 50% station footprint. If there is no overlap between the current network and the footprint of the station joining, the difference between the two network footprints is the same as the sum of the (50%) station footprint."

L360: 'relative underrepresentation' Does this refer to Fig 8b? Please add.

In general, there were some references missing in the result which have now been added (see also the following reviewer comments). Note that there are now separate maps for Germany and Europe (review comment further down), which makes for different numbering in some cases.

L362: Where do I see this? Compare 4b to 8b?

L365f: 'Figure 8 shows ... that great monitoring potential ...' But Fig 8 does not show monitoring potential, just footprint. Better refer to Fig. 6 b and d here.

L367f: Same as above: Where do I see this? Country contributions in Fig 8b versus 4b?

L385: Unclear: which scale do you refer to? How does a scale target a country? Scale of what? Please try to rephrase.

The reviewer is referring to:

"To plan for network expansion in relatively well-monitored countries, the scale should target a specific country, or even specific region within large countries."

We agree and have updated that it can be phrased better and have updated the text:

To plan for network expansion in relatively well-monitored countries, the analyses should target a specific country, or even a specific region within a large country.

L410f: Please give references to the specific QND studies you are referring to.

Reference to a specific study using reduction of uncertainty of underlying carbon fluxes as a metric for potential of station locations has been added:

However, normally the metric for considering potential station location in QND studies, such as previously mentioned Nickless et al. (2020), is reduction in uncertainty of underlying carbon fluxes and tend to cluster around the station like footprints.

L423: See above. Larger sensitivity to biogenic fluxes not proven.

We are more careful in how we phrase our conclusions:

"The stations pick up signals throughout the heterogeneous European flux landscape and show a large variation of sensitivities, with generally larger sensing capacity for biogenic fluxes than for anthropogenic emissions during the study period of summer 2020. The summer is most interesting

from a biosphere monitoring perspective and 2020 has also proven robust for longer time-periods in terms of our conclusions about the network."

**Technical comments**

L110 and elsewhere: 'Sect. 3.1' instead of just '3.1.

This has been implemented.

Updated equations in accordance with the point made above (L185ff).

We consider the model domain 15°W to 35°E and 33°N-73°N:

For a given country or region, $C$, we can look at $C_{i,j}$ which is the fraction of the country/region in a given grid cell $(i, j)$. We consider specific land cover types, $LC$, and use $LC_{i,j}$ which is the fraction of land cover within a given grid cell.

We establish the network view ($N_{i,j}(T)$) and equal view ($NEQ_{i,j}(T)$) of the flux land scape ($GEE_{i,j}(t_k)$): For each grid cell $i, j$ and hour $t_k$ leading up to when the air arrives at the receptor (T) where hour $t_k = t_1, t_2, t_3, \ldots, t_{240}$ (here $t_{240}$ is T, the time the air arrives at the receptors of the stations in the network).

$$N_{i,j}(T) = C_{i,j} \cdot LC_{i,j} \sum_{k=1}^{240} NFP_{i,j}(t_k) \cdot |GEE_{i,j}(t_k)| \tag{1}$$

$$NEQ_{i,j}(T) = C_{i,j} \cdot LC_{i,j} \sum_{k=1}^{240} \frac{\sum_{m,n} C_{m,n} \cdot NFP_{m,n}(t_k)}{\sum_{m,n} C_{m,n}} \cdot |GEE_{i,j}(t_k)| \tag{2}$$

Where $C$ and $LC$ are the fractional country grid of selected country or region. $NFP$ is the network footprint (see Sect. 2.4) and $GEE$ is the flux map and these change with time ($t_k$). $m$ and $n$ are coordinates of the grid cell.

The relative flux representation ($REP(T)$, used in Fig. 5b and Fig. 7b) is the ratio between the total sensing within the grid cells of the network view ($N_{i,j}(T)$) and the equal view $NEQ_{i,j}(T)$).

$$REP(T) = \frac{\sum_{m,n} N_{m,n}(T)}{\sum_{m,n} NEQ_{m,n}(T)} \tag{3}$$

The relative monitoring potential maps ($MP(T)$, used in Sect. 3.2 and Sect. 3.3) show the difference in the sensing between the network view and the equal view within the individual grid cells of the model.

$$MP(T) = \max\left(NEQ_{i,j}(T) - N_{i,j}(T), 0\right) \tag{4}$$

Cells where the equal view ($NEQ_{i,j}$) is greater, meaning sees more uptake, than the network view ($N_{i,j}$) will have positive values in the relative monitoring potential map ($MP$), and only these are displayed. Monitoring potential becomes especially high in areas where the current network is relatively blind, and the activity of the specific flux is relatively high. For the monitoring potential maps for the extended network, the equal view ($NEQ_{i,j}$) is kept the same as for the current network to facilitate effective comparison between the maps.

---

## Author Comment (AC2)

**RC2: 'Comment on acp-2022-756', Anonymous Referee #2, 31 Jan 2023**

*This is a nice paper describing the ecosystem surface influence of the ICOS network and the increased coverage with the addition of new sites. I recommend publication with some minor revisions described below.*

We thank the reviewer for their assessment of our manuscript. The individual points made are addressed below. Please note that the line numbers in the reviewer comments refer to the first version of the paper and our responses to the latest version of the paper.

**Minor comments**

*The abstract is overly long and not concise enough. Please condense it.*

The abstract has been shortened and now reads:

"The ICOS (Integrated Carbon Observation System) network of atmospheric measurement stations produces standardized data on greenhouse gas concentrations at 36 stations in 14 different European countries (November 2022). The placement of instruments on tall towers and mountains makes for large influence regions (concentration footprints). The combined footprints for all the individual stations create a "lens" through which the network sees the European $CO_2$ flux landscape. In this study, we summarize this view using quantitative metrics of the fluxes seen by individual stations, and by the current and future ICOS network. Results are presented both from a country-level and pan-European perspective, using open-source tools that we make available through the ICOS Carbon Portal. We target anthropogenic emissions from various sectors, as well as the land-cover types found over Europe and their spatiotemporally varying fluxes. This recognizes different interests of different ICOS stakeholders. We specifically introduce "monitoring potential maps" to identify which regions have a relative underrepresentation of biospheric fluxes. This potential changes with the introduction of new stations, which we investigate for the planned ICOS expansion with 19 stations over the next few years.

We find that the ICOS network has limited sensitivity to anthropogenic fluxes, as was intended in the current design. Its representation of biospheric fluxes follows the fractional representation of land-cover and is generally well-balanced considering the pan-European view. Exceptions include representation of grass & shrubland and broadleaf forests which is abundant in southeastern European countries, particularly Croatia and Serbia. On country scale the representation shows larger imbalances, even within relatively densely monitored countries. The flexibility to consider both individual ecosystems, countries, or their integrals across Europe demonstrates the usefulness of our analyses and can readily be re-produced for any network configuration within Europe."

*line 50: atmospheric needs to be mentioned here. You have mentioned land and ocean but the rest of the co2 is contributing to the atmospheric growth rate.*

The reviewer is referring to this section which has been updated (underlined part):

Up until now, about half of the $CO_2$ humans have emitted has been taken up by land (29% of total $CO_2$ emissions 2011-2020, Friedlingstein et al., 2022) or stored in the deep ocean (26% of total $CO_2$ emissions 2011-2020, Friedlingstein et al., 2022). The other half of the anthropogenic $CO_2$ remains in the atmosphere and contributes to the atmospheric growth rate which is 2.5 ppm for 2022 according to a preliminary estimate by Friedlingstein et al. (2022). On global scale, our common $CO_2$ trajectory will greatly depend on the capacity for storage in these carbon reservoirs and is important to plan our efforts under the Paris agreement. Furthermore, understanding the natural carbon exchanges between carbon reservoirs is important for our ability to track and verify changes in emissions (Balsamo et al., 2021).

Line 60. Full stop missing.

A full stop has been added.

Line 93. While the exact approach used here is not awfully common, there have been a few previous studies that have examined ecosystem representation of various networks with a slightly different tool (eddy flux) and should be mentioned. Malone et al., 2022 (https://doi.org/10.5194/bg-19-2507-2022) examined the ecological representation of the NEON network of eddy flux towers over the US but for the eddy flux footprints, which are much smaller than the concentration footprints / surface influence of the ICOS network. Pallandt et al., 2022 (https://doi.org/10.5194/bg-19-559-2022) also looked at ecosystem representation in the Arctic ecosystems.

We thank the reviewer for pointing us to these interesting references which indeed should be mentioned in our study. The following was added:

"For ecosystem sites, where fluxes rather than concentrations are measured, the "flux footprints" are small with influence mainly from the site's immediate surrounding. In this context, the idea of representation has been applied in Pallandt et al. (2022) to assess what Arctic ecosystem types are at the site locations compared to what ecosystems are found in the Arctic. Malone et al. (2022) similarly identified gaps in the U.S. NEON network based on representation of different clusters identified based on their ecological properties. In both studies, the evaluation of the network representation was subsequently used to advice on future expansion and how to evaluate the appropriateness of upscaling of the fluxes to larger regions."

Line 116. Do you mean STILT v2 (Fasoli et al.,2018, https://www.geosci-modeldev.net/11/2813/2018/). If not, I'd recommend upgrading. It's much faster to produce the footprints than v1.

We thank the reviewer for the suggestion. However, STILT was set up in the ICOS Carbon Portal in collaboration with C. Gerbig at MPI for Biogeochemistry and we share input data processing. Therefore, we use the STILT version that is supported by MPI-BCG.

Line 124. I don't understand this statement. If you want to combine the footprints with biogenic CO2 estimates, then you need to have at least 3 hourly (hourly would be better) time-step footprints back over the 10 days or you won't be able to interpret the CO2 signals. There is a strong diurnal cycle in CO2 flux from the biosphere with uptake during the day and emission at night. So you can't just take a 10 day footprint and multiply it by the net CO2 from VPRM. That's not the same as combining it in a sub-diurnal way. See Commane et al, 2017, Schiferl et al., 2022 for examples. It sounds like on Line 137 that you are using the time varying footprints for the interpretation so maybe I am just misunderstanding. In that case, maybe a little re-wording/clarification would be good. Did you combine the GPP and Resp from VPRM separately? In that case, did you use a static average number for the previous 10 days? I think that would actually be ok, given what you are doing with this study. But it needs to be explained.

We thank the reviewer for this important point and useful references. We did indeed multiply time-step aggregated footprints back over the ten days with static $CO_2$ fluxes from VPRM, but from GEE and Respiration rather than the net $CO_2$. We used the flux maps matching the time the air was modelled to arrive at the receptor, which indeed meant that we missed the strong diurnal cycle of especially GEE. To do better justice to this variation, we have updated our methods:

We have re-created the study with hourly time-step footprint back over the ten days, as pointed out as preferable by the reviewer. It means updated results throughout the study, but it did not change our main conclusions (see comparison below).

Hourly backward footprints required to re-create the study were created and saved specifically for our purpose and are not normally saved when the Carbon Portal Footprint Tool is run. This, in

combination with the computational resources required for the updated approach, means we cannot offer this to the general users of our online tool hosted at the Carbon Portal JupyterHub. Rather, we will update the tool to use a static average number for the previous ten days as suggested as an acceptable alternative by the reviewer.

The different approaches to arrive at network representation of different ecosystem fluxes are compared below for the current network year 2020 (same information as in paper Fig. 5b, and Fig. 8b for Germany, upper bars). The fair agreement between using the static 10-day average compared to the best approach now used for the paper gives confidence that it is feasible to use in the online tool.

To better understand the differences between the approaches, we have further examined the representation of coniferous forests; the higher overrepresentation in the static tool comes mainly from the night-time hours (0:00, 3:00, 6:00, 21:00) with average overrepresentations of the individual time-steps ranging from 12-24% compared to the "equal view". Our explanation for this is that there is essentially no GEE during these hours when backward resolved footprints are used, whereas the static 10-day average GEE mean activity also during these night-hours. Footprints representing the night, where there is less mixing of the air, will generally have more local influence on the measurements. Many of the ICOS stations, such as Hyltemossa, is close to/in coniferous forests which therefore explains the relative over-representation. This again stresses the advantage with the new approach suggested by the reviewer.

| Country | Total GEE | Broad leaf forest | Coniferous forest | Mixed forest | Other | Grass & shrub-land | Cropland | Pasture | Urban |
|---|---|---|---|---|---|---|---|---|---|
| Europe GEE "closest match" (preprint, outdated) | 3,12 | -12,56 | 8,88 | -3,72 | 56,83 | -52,93 | 8,60 | 7,21 | 59,45 |
| Europe GEE "static 10 days" (online tool) | 3,51 | -16,84 | 12,64 | -4,39 | 74,09 | -53,13 | 11,54 | -1,76 | 62,70 |
| Europe GEE "backward resolved" (paper) | 2,96 | -16,51 | 0,93 | -0,35 | 40,67 | -53,66 | 13,56 | 7,68 | 63,89 |

**Table 1. The over- (+) or under (-) representation of the current ICOS network within Europe (see paper Sect. 2.5). Results are given for GEE summer year 2020 with the different approaches to combine footprints with fluxes.**

The difference in results presented above will be shared in the online tool to inform users about how the methods are different and what this may mean for their results.

Line 178. Others.

"Other" changed to others.

The Results and discussion could be combined. I think it would read better and there is a lot of discussion in the Results section already. But that's just a suggestion.

We thank the reviewer for this suggestion but have decided to keep the sections separately. we have removed text that appears in both sections in a couple of places to avoid redundancy.

---

## Author Response (AR1)

We thank the reviewers for their assessment of our manuscript. The individual points made by reviewer one, followed by reviewer two, are listed and addressed below. Please note that the line numbers in the reviewer comments refer to the preprint.

**Major comments reviewer one.**

Comparison to real world data:

The study is solely based on bottom-up anthropogenic, oceanic and biogenic $CO_2$ fluxes combined with atmospheric transport simulations. Table 1 presents mean model simulated contributions to $CO_2$ mole fractions at the different sites of the ICOS network, but there is no validation of how well these mean mole fractions compare to observed mole fractions. Even more interesting would be an indication of how well the simulations fit the individual afternoon observations. Demonstrating the general ability of the model to capture the observed mole fractions for one or a few sites would strengthen the credibility in the model system and all following analysis and conclusions on the network. If such comparisons already exist for the presented model system, it would be sufficient to refer to these results. Otherwise, I would suggest to include a comparison for at least the site that is already discussed as an example, Hyltemossa.

We agree with the reviewer that this type of comparison would be useful and strengthen the credibility of the modelling system. We have considered the agreement between modelled and measured concentrations for all stations in the 2022 ICOS atmospheric station network, and the results are shown in the table below. For year 2020, the correlation coefficient (r) is on average 0.82. The reviewer expressed interest in the afternoon hours, and if only these (12:00 and 15:00 UTC) are considered, the average correlation coefficient is rather 0.86. We choose to mention the correlation given all hours of the day we have model results for (0:00, 3:00, 6:00, 9:00, 12:00, 15:00, 18:00, 21:00) as we do not exclude any hours in our study. However, we include both selections in Appendix B:

| STILT station | Sampling height | r (all hours) | r (12, 15 CET) | Bias (all hours) | Bias (12, 15 CET) | RMSE (all hours) | RMSE (12, 15CET) |
|---|---|---|---|---|---|---|---|
| BIR075 | 75 | 0.88 | 0.92 | -2.66 | -1.84 | 4.32 | 3.69 |
| CMN760 | 8 | 0.88 | 0.90 | -0.72 | -0.20 | 3.53 | 3.54 |
| GAT344 | 341 | 0.88 | 0.89 | -1.52 | -1.36 | 4.56 | 4.43 |
| HEL110 | 110 | 0.91 | 0.92 | -4.39 | -4.25 | 7.01 | 6.77 |
| HPB131 | 131 | 0.77 | 0.79 | -2.03 | -1.39 | 6.24 | 6.01 |
| HTM150 | 150 | 0.90 | 0.93 | -2.10 | -1.07 | 4.36 | 3.49 |
| IPR100 | 100 | 0.70 | 0.85 | -10.45 | -3.89 | 15.46 | 8.97 |
| JFJ | 5 | 0.81 | 0.81 | -1.21 | -1.23 | 4.10 | 4.24 |
| JUE120 | 120 | 0.57 | 0.72 | 0.82 | 3.64 | 12.77 | 11.62 |
| KIT200 | 200 | 0.68 | 0.82 | -7.05 | -4.28 | 13.22 | 9.97 |
| KRE125 | 125 | 0.79 | 0.86 | -3.25 | -1.48 | 6.60 | 4.94 |
| LIN099 | 98 | 0.71 | 0.86 | -3.22 | -1.50 | 8.82 | 5.33 |
| LMP | 8 | 0.92 | 0.92 | -0.76 | -0.63 | 2.25 | 2.21 |
| NOR100 | 100 | 0.84 | 0.92 | -3.15 | -1.27 | 5.65 | 3.73 |
| OPE120 | 120 | 0.82 | 0.87 | -1.93 | -0.90 | 4.97 | 3.89 |
| OXK163 | 163 | 0.77 | 0.75 | -1.10 | -1.03 | 5.49 | 6.12 |
| PAL | 12 | 0.89 | 0.95 | -0.69 | -0.71 | 3.53 | 2.82 |
| PRS | 10 | 0.84 | 0.87 | -0.60 | -0.18 | 3.41 | 2.79 |
| PUI084 | 84 | 0.89 | 0.95 | -2.27 | -0.91 | 4.64 | 3.26 |
| PUY | 10 | 0.83 | 0.86 | -0.88 | 0.25 | 3.44 | 3.06 |
| SAC100 | 100 | 0.64 | 0.83 | -1.60 | 0.20 | 9.90 | 6.26 |
| SMR125 | 125 | 0.92 | 0.95 | -2.21 | -1.15 | 4.13 | 3.14 |
| STE252 | 252 | 0.84 | 0.88 | -2.85 | -2.12 | 6.34 | 5.53 |
| SVB150 | 150 | 0.92 | 0.95 | -2.07 | -1.37 | 3.77 | 3.11 |

| | | | | | | | |
|---|---|---|---|---|---|---|---|
| TOH147 | 147 | 0.83 | 0.83 | 0.17 | 0.01 | 4.72 | 4.82 |
| TRN180 | 180 | 0.79 | 0.86 | -1.98 | -0.88 | 5.01 | 3.83 |
| UTO | 57 | 0.91 | 0.90 | -1.65 | -1.75 | 3.71 | 3.85 |
| WAO | 10 | 0.70 | 0.60 | -3.03 | -3.31 | 8.30 | 10.66 |
| ZSF | 3 | 0.87 | 0.88 | -0.90 | -0.69 | 3.65 | 3.55 |

**Table A1. Agreement between modelled and measured CO₂ concentrations at stations in the current ICOS network year 2020. Results given all hours with model results (0:00. 3:00. 6:00. 9:00. 12:00. 15:00. 18:00 and 21:00 CET) and only afternoon hours (12:00 and 15:00 CET) are presented. A negative bias means model underestimation.**

[Figure]

**Figure 1. Timeseries for Hyltemossa (150 m) of modeled CO₂ concentrations from the STILT Carbon Portal implementation and measured CO₂ concentrations year 2020. All hours with model results are considered; 0:00, 3:00, 6:00, 9:00, 12:00, 15:00, 18:00, 21:00.**

This information has been added to Sect. 2.2 of the paper:

"The total modelled concentrations can in turn be compared to measured concentrations to assess the performance of the modelling system; with an average correlation coefficient of 0.82 for all stations in the current network year 2020 there is generally a good agreement. The statistics of model vs. observation comparison for the individual stations can be found in Table A1 (Appendix A)."

Respiration fluxes:

A major part of the discussion is focused on the network characterisation with respect to summertime gross primary production (or here gross ecosystem exchange, GEE).

However, ignoring respiration for the network characterisation seems like abandoning the study half-way. Respiration fluxes show a distinctly different temporal evolution as compared to GEE fluxes. Relative contributions from different land cover types may differ between respiration and GEE. Average footprints differ between summer and winter (as shown in Fig. 2). All these points indicate that conclusions drawn in terms of representatively monitoring GEE do not necessarily apply to respiration or NEE. This is already demonstrated by the flux contributions at the site Hyltemossa (Fig. 3), where contributions by land cover differ strongly between GEE and respiration as well as between summer and winter. If the aim of the ICOS atmospheric network is the observation of the complete carbon budget and not just the summertime CO2 uptake, the present analysis should integrate a discussion of respiration and/or NEE.

We recognize the reviewers concern with our focus on GEE and specifically the summer. Tied to other comments by the reviewer, we have made this clearer and state that the conclusions about the network are with regards to this. During the study, we considered the inclusion of respiration but decided to proceed with only GEE and provide our rationale for this below. However, we will make it possible to choose between respiration and GEE in our online tool in its next release.

1. In terms of NEE, this is what we state in the paper (Sect. 2.5).

"We use GEE (Gross Ecosystem Exchange) rather than NEE (Net Ecosystem Exchange = GEE + Respiration) as footprint-weight to prevent nearly cancelling out photosynthesis and respiration signals to influence the network view. The GEE-view thus highlights areas with high biogenic activity which are especially important to monitor; high activity generally means greater uncertainties in the current estimates and higher potential for long-term carbon storage." So recognizing that NEE is small and changes sign, we are left with the choice of considering GEE or Respiration as proxy for ecosystem sensitivity of the network.

2. Our focus on GEE, and not respiration.

The large signals associated with different land cover types that have high GEE values (large sink) in general also have high respiration values (large source). This is expected, as respiration is a function of carbon in soils and litter, which is large for highly active vegetation (high GEE). Correlation coefficients for both, associated with different land cover types, range from 0.74 for pasture to 0.96 for coniferous forest. Note that the different diurnal cycle in respiration compared to GEE cancels out (see our signals in Table 1) in the average over the whole time-period (summer 2020).

The correlation between high GEE and respiration is also evident spatially when we evaluate how different our relative monitoring potential maps would look like if respiration would be used instead of GEE: For Europe, all the land cover classes, the correlation coefficient is over 0.8. A general pattern in the maps is that the monitoring potential is slightly more spread out when it comes to respiration. This is because of the relative nature of the monitoring potential maps, in which the most active locations (in GEE especially forests) get relatively more weight. This is as intended, as highly active vegetation should receive extra attention because of their influence on the final carbon balance.

| | Broadleaf forest | Coniferous forest | Mixed forest | Other | Grass and shrubland | Cropland | Pastures | Urban |
|---|---|---|---|---|---|---|---|---|
| GEE summer 2020 | -16,5 | 0,9 | -0,4 | 40,7 | -53,7 | 13,6 | 7,7 | 63,9 |
| RESP summer 2020 | -8,9 | 5,3 | -8,4 | 50,2 | -53,6 | 7,3 | -2,4 | 62,0 |

**Table 2. The over- (+) or under (-) representation of the current ICOS network within Europe (see paper Sect. 2.5). Results given for respiration and GEE summer year 2020.**

3. Our focus on summer, rather than the whole year

The reason we focus on summer is because it is the time of the year with highest biogenic activity and when it is most important that we are able to monitor the ecosystems. Moreover, this is the gross flux that dominates during daytime when the data is used for inverse modelling. However, we see the value in more carefully consider the sensitivity of the results to the use of a different time-period.

| | Broadleaf forest | Coniferous forest | Mixed forest | Other | Grass and shrubland | Cropland | Pastures | Urban |
|---|---|---|---|---|---|---|---|---|
| GEE summer 2018 | -5,7 | 2,8 | -3,9 | 70,3 | -54,1 | 4,7 | 12,6 | 72,8 |
| GEE summer 2019 | -1,7 | 5,1 | 0,8 | 70,8 | -50,5 | 12,2 | 9,6 | 71,2 |
| GEE summer 2020 | -12,6 | 8,9 | -3,7 | 56,8 | -52,9 | 8,6 | 7,2 | 59,5 |
| GEE summer 2020 backward* | -16,5 | 0,9 | -0,4 | 40,7 | -53,7 | 13,6 | 7,7 | 63,9 |
| GEE summer 2021 | -9,0 | 12,2 | -2,6 | 63,2 | -51,4 | 13,4 | 12,8 | 65,4 |

**Table 3. The over- (+) or under (-) representation of the current ICOS network within Europe (see paper Sect. 2.5) for summers years 2018-2021. *Updated approach which is computationally heavier, and we only have results for year 2020 currently. See replies to RC2 for more details**.

We have added a discussion about representativeness of summer 2020 in Sect. 4 of the paper:

"For the years 2018-2021 we found a similar representation of the different land cover types as in summer 2020. Exceptions include less pronounced underrepresentation of broadleaf forest fluxes and larger overrepresentation of coniferous forest fluxes on European scale. For Germany, the results are even closer with the most notable difference in the underrepresentation of mixed forests. In the Carbon Portal online tool an analysis with more

computationally efficient time-step aggregated footprints is offered and this approach was used for the other years (see Sect. 6). The differences to the approach used in this study are also discussed in more detail there. "

**Minor comments by reviewer one.**

L12, L75: When introducing the footprints for the first time, it may be helpful to call them 'concentration footprints' to distinguish them from 'flux footprints' as used in the eddy covariance community.

The suggestion was implemented.

L13: 'European flux landscape'. Not very specific. Better 'carbon flux' or even 'CO2 flux'.

"European flux landscape" changed to European $CO_2$ flux landscape.

L16: Same as previous: 'anthropogenic CO2 emissions'. In general, a note in the conclusions could be added that the results obtained here are exclusively valid for the CO2 flux landscape. Other parameters observed by the network (CH4, N2O) exhibit a completely different flux distribution and a similar analysis could be carried out in order to characterise their gaps and monitoring potentials.

The suggestion was implemented.

We decided to stress the validity of the study for the $CO_2$ flux landscape already in the introduction:

"We chose in this study to quantify and summarize the capacity of the ICOS atmospheric observing network to sense the underlying $CO_2$ flux landscape. Other mole fractions observed by the network including $CH_4$ and $N_2O$ exhibit different flux distributions and would therefore need separate analyses to characterise their gaps and monitoring potentials. "

Fig 1 (and others): Although the spatial reference system is noted in the legend, their are no axis labels which would allow for geographic referencing. Please add.

The suggestion was implemented for all maps.

In caption, please mention source of land cover classes given in the figure.

The reference to the dataset was added.

L57: Be more specific. What kind of measurements?

The text has been updated and reads:

"Measurements of atmospheric mole fractions have traditionally been collected at remote islands, mountain tops, or other locations at large distance from direct emissions or uptake, to find well-mixed conditions that represent background atmospheric levels (Conway et al., 1994)."

L66: European climate zones stretch form sub-tropical (dry summers) through temperate (Central Europe) to sub-arctic. Please rephrase accordingly.

The text is in accordance with the map in our reference (Beck et al., 2018). However, we realize that "temperate with dry and hot summers" sound a bit unfamiliar and now use "Mediterranean with dry and hot summers" instead:

Europe has multiple climate zones ranging from Mediterranean in the South with according to the Köppen-Geiger classification dry and hot summers, through temperate, to cold Northern sub-Arctic climate without a dry season (Beck et al., 2018).

L69: Replace 'has more' by 'dominated by'.

The suggestion was implemented.

L95: There is no absolute measure of sensing capacity, I would argue that the metrics introduced here should be called 'more quantitative' instead of 'quantitative'. Or at least add that the introduced metric remains semi-objective as you could have used a slightly different metric instead.

We agree and have implemented the suggestion to call it "more quantitative" instead of quantitative.

L107f: Please refer to the 'Code availability' section here. I tried to follow the links and get the jupyter notebook to run. However, I got a 'page not found' error when trying the link for Storm 2022 in the list of references. Hence, I was not able to check whether there are working tools behind this or not. In the ideal case, getting your tools to run should not cost a whole day to set it up.

Unfortunately, the link in the original PDF also included the page number of the paper, which meant clicking it did not work. The preprint and new version of the paper have working links. We are sorry about this.

Furthermore, an up to date version of the tool is also available on the Carbon Portal JupyterHub (https://exploredata.icos-cp.eu/): login with email address and current password "francis" (the update may change, see instructions here: https://www.icos-cp.eu/data-services/tools/jupyter-notebook/exploredata-password). If the JupyterHub is used, there is no setup of the tool required and users can start their runs immediately.

Sect. 2.1: In this section it does not become clear for which period the analysis was done. Later it is mentioned that the analysis was done for summer (winter?) 2020. How representative is a single summer? Was the summer 2020 rather typical are characterised by extremes?

We have added the following text to the end of section 2.1:

"Footprints for summer (JJA) were used for the subsequent analyses. Additionally, footprints for winter (DJF) year 2020 were used only in the analysis of individual stations exemplified with Hyltemossa (Sect. 3.1)."

In terms of the representativeness of a single summer, please see the response to the respiration discussion (major comment number two). There are naturally differences between summers, and we are more carefully stressing that the conclusions are specifically for summer 2020 in the updated version of the paper.

L122: What was the spatial resolution of ECMWF data used?

We have added this information to the paper:

"Meteorological conditions drive the transport and are represented by three-hourly operational ECMWF-IFS analysis/forecasts at 0.25 degrees resolution."

Sect 2.2: Maybe change title so that it includes the description of the utilised fluxes, for example "CO2 fluxes and simulated signals at the stations"

The suggestion was implemented.

L130: This is a rather old version of EDGAR. Why is it used instead of the more previous releases?

We use the Carbon Portal Footprint tool which is indeed based on an older version of EDGAR. However, the emissions have been extrapolated to 2020 using energy statistics and the dataset has also been temporally disaggregated to hourly resolution. For more detailed information, readers can consult the description of the set-up which is included as a reference in the paper:

Karstens, U (2022): https://meta.icos-cp.eu/objects/XX3nZE3l0ODO9QA-T9gqI0GU.

L135ff: Not quite clear how this was done? Was VPRM run twice (once with SYNMAP and once with HILDA land-cover) or simply on SYNMAP and the remaining land-cover analysis was done with HILDA only? How does resolution of these two land cover datasets relate to the output resolution of STILT? Where is SYNMAP data published? In a quick search I could only find the paper but no link to the data (as claimed in the data availability section).

We realize how this needs to be better explained and have added to the text (see below). In short, the VPRM is only run once, and we use the output published at the Carbon Portal (https://doi.org/10.18160/VX78-HVA1). The creators of the VPRM biosphere fluxes refer, like we do, to the paper by Jung et al., 2006:

Gerbig, C.: Parameters for the Vegetation Photosynthesis and Respiration Model VPRM (Version 1.1), ICOS-ERIC – Carbon Portal [data set], https://doi.org/10.18160/R9X0-BW7T, 2021a.

SYNMAP has been accessed upon a request to the authors (M. Jung). The Vegetation fraction maps for the STILT resolution (1/8 x 1/12 degrees) were created based on aggregated groups of the 1km resolution original SYNAMP

land cover. To again disaggregate the land cover, we alternatively re-create the timeseries using HILDA because we see advantages with the dataset compared to using the original SYNMAP (see details below). The resolution of the HILDA land cover is also 1km, and in the same way as the creators of the VPRM fluxes, we prepared alternative vegetation fraction maps for the combination with STILT footprints.

Added to Sect. 2.2:

"The biosphere-flux derived signals computed during the footprint calculation are attributed to groups of aggregated SYNMAP (Jung et al., 2006) land cover categories, which is also the map used to parameterize the VPRM model. The aggregation used within the STILT Footprint Tool makes for broad categories such as "crop and tree", which is attributed to over half of the land area in the model domain. To again disaggregate the land cover, we alternatively re-create the timeseries of biogenic signals by combining the footprints for hourly backward time-steps with the temporally resolved flux maps but attribute the resulting biogenic signals to the different land cover categories in HILDA (Winkler et al., 2020). The HILDA land cover is a synthesis product built on multiple heterogenous datasets including several satellite data products which were published after SYNMAP was created (Winkler et al., 2021). Another advantage is that the HILDA map used in this study represents year 2018 as opposed to the year 2000. "

L145: It could still be interesting if by choosing a lower inlet height one could focus the view on specific land types. However, I agree that by selecting the more convective afternoon situations, differences my not be too large as compared to the uppermost inlet height. Maybe something for future analysis.

Indeed, to focus on specific land cover types is something that could be the reason for choosing a different inlet height. We have added a note about this in the discussion (Sect. 4):

"In case of multiple inlet heights at a station, the highest level was selected because the top-level measurements have the largest footprints and are generally chosen to provide measurements for regional and global inverse modelling systems. By choosing a lower inlet height one could potentially focus the view on specific land types."

For the interest of the reviewer, we can share that we had a look at what the difference would be for Hyltemossa's lower inlet heights (70 m and 30 m) compared to the highest inlet (150 m) and found increasing shares of coniferous forests in the average footprints with decreasing inlet heights. The generally more local sensing associated with lower inlet heights agree well with this picture as coniferous forests is found in abundance in the area surrounding Hyltemossa.

L155: The 50 % are arbitrary. Why not use the complete footprint here to calculate the contributions from different land cover? How large would the difference be?

In the paper we provide the following rationale for using the 50% footprints:

"The hourly backward time-steps footprints for the individual stations in the network are reduced to grid cells with the highest sensitivity values that in combination add up to 50% of the sum of the hourly backward time-step footprints sensitivities. This follows the approaches of Henne et al. (2010) and Oney et al. (2015) and is intended to emphasize areas with significant local influence."

Furthermore, the uncertainties in the sensitivity values derived from the hypothetically dispersed particles increase further backward in time, when they are further from the station. This is reflected in a dynamic aggregation of cells to final footprints with large areas far from the station attributed with the same sensitivity values. For our study, this means generally small differences in using the 50% footprints compared to the full footprints as much of the sensitivity is spread out evenly over areas of diverse land cover. A sensitivity test was done with timestep-aggregated footprints and shows the following differences in representation:

| | Broadleaf forest | Coniferous forest | Mixed forest | Other | Grass and shrubland | Cropland | Pastures | Urban |
|---|---|---|---|---|---|---|---|---|
| GEE summer 2020 50% footprints | -16,5 | 0,9 | -0,4 | 40,7 | -53,7 | 13,6 | 7,7 | 63,9 |

| GEE summer 2020 100% footprints | -14,30 | 3,99 | -7,72 | 38,60 | -43,88 | 5,18 | -1,11 | 49,79 |

**Table 4. The over- (+) or under (-) representation of the current ICOS network within Europe (see paper Sect. 2.5). Results given for GEE summer year 2020 with 50% footprints and 100% footprints.**

L159: Why take the maximum instead of the sum? In the real world multiple sites would receive information from the grid cell, which adds to the information available to any inverse modelling approach. By using the maximum here, a general underestimation of the total network sensitivity results.

When footprints of stations overlap, we take the maximum sensitivity to the cell of the two stations. Rather than when taking the sum of the overlap, it focuses the analysis on representation of fluxes spatially. With our choice, placing a new station close to an old would not double the capacity to monitor fluxes. We agree that in a quantitative estimate of $CO_2$ fluxes using such footprints, one would indeed use the sum of sensitivities and derive a flux with smaller uncertainty under an overlapping footprint.

However, we found that when considering the sum instead of the maximum for our study, the differences are small because there is anyway limited overlap between the atmospheric stations in our network, as they are generally at quite a large distance from each other. We for example looked at how much new stations caused overlap with the "current" network:

"Overlap (see Sect. 2.4) with the sensing capacity of the current network is close to zero for Spanish and Irish stations with respect to their 50% summertime footprints, and well-below 10% for all other added stations with exceptions for German Schauinsland (SSL) and Polish mountain-station Sněžka (SNZ)."

L163: What is quantified here? Additional area covered? Differences in land cover contributions? Please clarify.

This is related to the last point and has been clarified in the text. What is quantified is the "loss" in network sensitivities from taking the maximum rather than the sum (last reviewer point):

"To estimate the overlap in sensitivity between a current network footprint and the 50% footprint of a station that is included in the extended network, the effect of its inclusion in an updated network footprint is analysed; the difference between the spatial sums of the two network footprint sensitivities is compared to the spatial sum of the 50% station footprint. If there is no overlap between the current network and the footprint of the station joining, the difference between the two network footprints is the same as the sum of the (50%) station footprint."

L168: Same question as above: Are VPRM fluxes based on HILDA?

No, VPRM fluxes are not based on HILDA. This is addressed in the reply to the question above (L135ff).

L169: The abbreviations 'GEE' and 'NEE' are never properly introduced.

We have added proper introductions to the abbreviations where they first appear.

L173: Also see major comment above: It would still be interesting to see if results would be similar for a respiration view which would need to include a whole year of simulations. As you say respiration is similarly important for the carbon budget and the network hence should equally be representative for respiration. Even for GEE it would seem beneficial to include a whole growing season and not just the summer as uptake will differ in timing between different land-cover, especially crops versus forest, and climate region, early growing season in the South limited by water availability in the summer versus northern climate zones with late onset of growing season. Only focusing on summer may introduce a bias here as well.

This was addressed in the reply to the major comment.

We encourage users to further test different time periods with our online tool.

L185ff: These equations need to be properly typeset and variables a to h defined as such. It is not clear over which dimensions the individual variables run (space, time, land cover) and over which of these the sums are defined.

We agree that this is needed to make it clearer and have updated the equations. They are provided at the end of this document.

L189: To me it is not really clear at this point why you need the footprints here. Is the area mean/total not simply defined by averaging/summing the proxy data over the area? Or do you additionally limit to the 50% footprint? Which I think would not make sense since you would limit the equal view already to what is within the direct view of the network. Or is the total simulated sensitivity redistributed equally to the target area?

The reviewer is referring to the section about the creation of the equal view footprint (Eq. 2 at the bottom of the document). With the updated text and the properly typeset variables we hope this is now clear. We try to also give the reviewer an answer here:

We combine "equal view network footprints", based on the original network footprints, with underlying data to arrive at what signals the network would pick up if the monitoring potential was spread out evenly. This feels intuitive to us as we can change what the network represents by consciously adding stations in areas with high monitoring potential of desired ecosystem types. However, this indeed gives the same results as using average proxy data in combination with the network footprints.

L195: Why is this called a mask? Isn't it simply the fraction in each grid cell. Potentially reaching from 0 to 100 %

It is indeed the fraction in each cell and the text has been updated in accordance.

L196: So e and f are obtained for each land cover type, correct? Please indicate this by a running index on those variables that are defined for different land cover. Supposedly c and d as well.

Yes, indeed. The equations are now properly typeset, and it is hopefully clear (provided at the end of this document).

L207: Got me all confused here. f > e would give negative h according to equation 4.

The absolute value of GEE was used. Hopefully it is all clear now with the updated equations (provided at the end of this document).

Table 1: Some columns are given with 1 others with 2 significant digits. For example the 'Residential' contribution is 0.0, 0.1, or 0.2 everywhere. Difficult to make out differences from this. Should GEE not be given with a negative sign to indicate uptake? I don't think this is ever clearly spelled out anywhere.

We have updated the table which now have two decimals and GEE is indicated as negative.

L222f, L233f: That conclusion is too general since only GEE is looked at. The picture would, most likely, look very different if NEE would be evaluated. Please consider that the CO2-only observations cannot distinguish between anthropogenic, uptake and respiration. Hence, only concluding from GEE that the network is mostly sensitive to biogenic is not valid. The conclusion may still be correct for summer (but even that is not mentioned here), but you would need to show with a NEE view!

Please see our response to the second major comment.

L255: These cites are not indicated in the figure. Please add for reference.

The reviewer is referring to Malmö and Copenhagen which have now been added to Fig. 2a.

L261ff: Is this analysis based on land cover alone (not fluxes)? Is the land cover weighted by footprint sensitivity before calculating the shares or is just the area analysed? Would be nice to add two bars for the total to Fig 2b (the analysis by direction) as the text seems to be discussing the total rather than any specific direction.

Yes, it is based on land cover alone and weighted by footprint sensitivity.

We have updated the legend text to make this clearer:

"summer (JJA) and winter (JFD) land cover shares weighed by the seasonal footprints split by direction."

The suggestion to add two bars for the summer and winter total has been implemented and updated graph looks like this:

[Figure]

The reviewer is referring to this paragraph:

"During the winter, biosphere respiration of $CO_2$ is almost as large signal as the anthropogenic contribution, which is dominated by energy production, followed by residential. Interestingly, emission sources within Sweden only contribute about 23% of the anthropogenic signal at the station. The remainder is transported from emission sources further away, emphasizing the importance of long-range transport."

Yes, this is true for Hyltemossa during winter 2020. This is a station located quite far north (lat: 56.1, lon: 13.4) and there is little biogenic activity during the winter. More interesting from a monitoring perspective when it comes to the biosphere is what is going on when the biosphere is active. We have made sure to clearly state that our analyses focus on the summer, when the biosphere signals are generally much larger than the anthropogenic signals. We also point to that sub-sampling of timeseries can be used to avoid measurements that are contaminated in subsequent modelling:

"However, it is important to remember that the signal averages include peaks in anthropogenic signals during particular hours especially when the wind transports air from large point source emitters. For example, the German station Jülich (JUE) is located only 10 km from a coal-fired power plant that accounts for about 4% of Germany's total emissions (E-PRTR, 2020). The average signal is 5.6 ppm but below 1.0 ppm about 20% of the time. Careful sub-sampling of time series, as suggested also by Oney et al. (2015), could allow for either avoiding anthropogenic influence or concentrating on its analysis."

Fig. 4: All the labels are way to small. Also the shading for the footprints is not necessarily well chosen and does not offer many details on the actual distribution as it mainly shows the location of the sites. It would also be interesting to discuss some of the apparent differences between sites. Some stick out with very large sensitivity in the surroundings, others are hardly visible.

We have increased the font of all figures.

We have discussed the way the footprints look and decided to keep the current color shading. An alternative would be to use a logarithmic scale which would lessen the emphasis on the areas around the station locations. However, we want to be careful with this not to overstate the network monitoring capacity.

In terms of the differences between stations, we have added the following:

"For stations with an air inlet close to the ground the sensitivity can be tenfold that of a mountain station. These large differences in sensitivities between stations are evident in the network maps; the southernmost station

Lampedusa with an inlet height of 8 meters (see Table B1) has strong local influences represented by saturated colors close to the station (Fig. 4a; Fig. 9a). Signals at such low-inlet stations are potentially larger and would in turn have greater influence on resulting monitoring potential maps."

L287f, Fig. 4b: I don't see the big difference for coniferous forests. I see differences for grass/shrub and also for urban, but the forests seem to be represented rather well. You could use percentages in the text to underline your point.

The difference in share of coniferous forests in Europe compared to the network is indeed not as large for coniferous forests as for the land cover classes grass & shrubland and urban. This is what figure 4b looks like (and Europe, furthest to the right, is discussed):

[Figure]

We decided against using percentages in the text because it gives the impression of a quantitative analysis: it is qualitative analysis, and the result will be dependent on the choices we have made.

The network share of coniferous forests is 20.6% compared to 17.8% within Europe. The difference for grass & shrubland is indeed larger with a network share of 4.5% compared to 10.3% within Europe.

The difference for the land cover type "urban" is larger than that for coniferous forest (9.7% network share compared to 5.9% within Europe), but we choose to focus on the non-urban land cover classes here. In the flux discussion we mention the overrepresentation of the urban land cover:

"The overrepresentation of fluxes associated with the land cover type "urban" (see Fig. 5b), despite the ICOS network targeting natural fluxes, is explained by the relatively high density of stations in western and central Europe; countries with highest sensitivity per area unit (Switzerland, the Netherlands, and Germany; see Fig. 4b) are also counties with some of the highest population densities."

Fig. 5: In b, do the upper and lower bars refer to current and extended network, respectively? Please add information to caption or plot.

The captions for the plots (also for Germany, Fig. 8) were updated:

"Figure 5: (a) Share of flux (GEE) per land cover within Europe compared to the network GEE-view for the current and extended ICOS network within Europe. (b) The over- (+) or under (-) representation of the current ICOS network (upper bars) compared to the extended (lower bars) ICOS network within Europe (see Sect. 2.5)."

L304f: Where do we see this? Is this in Fig 5?

We have updated the text to reflect where this can be seen:

The underrepresentation of broadleaf forest fluxes (see Fig. 5b), despite a fair LC-view (see Fig. 4b), means that broadleaf forests outside the focus of the network are more active than those currently sensed.

L312f: Could you please add references to the figures where the individual points can be seen? 'underrepresented flux' in Fig 5? 'Monitoring potential for Serbia and Croatia' in Fig 6c,d?

We have updated the text to reflect where this can be seen (Sect. 3.2.2):

"Grass & shrubland is the most underrepresented flux (see Fig. 5b) and shows greatest potential for monitoring in Serbia and Croatia (see Fig. 6b). Within the ICOS membership countries, Scandinavia has the highest potential (see Fig. 6b)."

Fig. 6: Are the same color scales used for the main plot and the German inset? Were these monitoring potentials for Europe and Germany calculated separately. Why would they look so different in the main plot and the inset? What are the actual values (not indicated on the color scale)? How should these be interpreted? I think it would be easier to understand if you don't present the European and German case in one plot, but have two separate plots.

We put the maps for Germany and Europe in separate map because they indeed have their relative monitoring potentials calculated separately. We see how it is confusing especially as Germany is white in the maps showing Europe and have colors in the insert map. To clarify how this can be, we have added a short explanation in the caption for the figure with the four maps showing Germany (Fig. 7):

"Same as Fig. 6, but relative monitoring potential within Germany as opposed to Europe. In the European context, Germany is well-monitored and appears to have no relative monitoring potential (white)."

We choose not to use actual values because the "equal view" and the "network view" of the fluxes used to establish the GEE-view is based on network footprints; the resulting values from combining the sensitivities of multiple stations (network footprints) with fluxes only have meaning in the relative sense. How the "equal view" and "network view" is used to create monitoring potential maps can be found in the updated methods section.

L322: Unclear. Do you mean that because of large local fluxes there is little additional signal for regions farther away? Even with the next sentence this is somewhat unclear.

The reviewer is referring to the following text (underlined):

"The overrepresentation of sensing of fluxes associated with the land cover type urban, despite the ICOS network targeting natural fluxes, is explained by the relatively high density of stations in central Europe; countries with highest sensitivity per area unit (Switzerland, the Netherlands, and Germany) are also counties with some of the highest population densities. The relatively high sensing per unit area in central Europe also means that these countries show little or no monitoring potential on European scale. This should not be interpreted as if their monitoring is "complete"; it only means they are well-monitored relative to other areas in Europe. For expansion of national networks, the same approach can be employed on country scale to analyse flux representation and highlight relative monitoring potential which we will illustrate for Germany."

The confusion regarding the concept of monitoring potential is evident also from the last point the reviewer made and we are thankful to have the chance to clarify this. The changes in response to the last point, and the updated equation with proper typeset variable (provided at the end of this document) will hopefully help, and we have updated the text the review is referring to:

"The high sensitivity of the network within these countries compared to the rest of Europe also means that their fluxes are relatively well-monitored and appear low in monitoring potential when all of Europe is considered (see Eq. 1-Eq. 4). This should not be interpreted as if their monitoring is "complete" and for expansions of national networks it is advisable to consider the relative sensing within the individual countries which we will illustrate for Germany."

L336: There is no table 3 in the manuscript. Only the single Table 1! Did you mean that? But how do I see relative contributions there?

Yes, we refer to Table 1 and not the non-existing Table 3. The text has been updated in accordance.

Figure 8: Do the two different color shadings represent the same value range? Please clarify by either adding a second color palette or clarify in the caption.

Note that with the updated maps for Germany, this is now Figure 9.

We see how this is confusing and have clarified the different shadings in the caption:

Figure 9: (a) Summer (JJA) 2020 extended network footprint overlaid on the current network footprint. The same level of color saturation of green and blue have the same meaning.

We chose not to add a second color palette as the focus is on the extended network which has the shown palette. The shading for the current network is the same as in the figure where it is shown separately (Fig. 4a).

L352: What does 'overlap' mean here? Is this with respect to the cropped 50 % footprints?

The reviewer is referring to this section which we have underlined with the updates to make the meaning of "overlap" clearer:

"Overlap (see Sect. 2.4) with the sensing capacity of the current network is close to zero for Spanish and Irish stations with respect to their 50% summertime footprints, and well-below 10% for all other added stations with exceptions for German Schauinsland (SSL) and Polish mountain-station Sněžka (SNZ)."

The updated description in section 2.4:

"To estimate the overlap in sensitivities between a current network footprint and the 50% footprint of a station that is included in the extended network, the effect of its inclusion in an updated network footprint is analysed; the difference between the spatial sums of the two network footprint sensitivities is compared to the spatial sum of the 50% station footprint. If there is no overlap between the current network and the footprint of the station joining, the difference between the two network footprints is the same as the sum of the (50%) station footprint."

L360: 'relative underrepresentation' Does this refer to Fig 8b? Please add.

In general, there were some references missing in the result which have now been added (see also the following reviewer comments). Note that there are now separate maps for Germany and Europe (review comment higher up), which makes for different numbering in some cases.

L362: Where do I see this? Compare 4b to 8b?

L365f: 'Figure 8 shows ... that great monitoring potential ...' But Fig 8 does not show monitoring potential, just footprint. Better refer to Fig. 6 b and d here.

L367f: Same as above: Where do I see this? Country contributions in Fig 8b versus 4b?

L385: Unclear: which scale do you refer to? How does a scale target a country? Scale of what? Please try to rephrase.

The reviewer is referring to:

"To plan for network expansion in relatively well-monitored countries, the scale should target a specific country, or even specific region within large countries."

We agree and have updated that it can be phrased better and have updated the text:

To plan for network expansion in relatively well-monitored countries, the analyses should target a specific country, or even a specific region.

L410f: Please give references to the specific QND studies you are referring to.

The reference to a specific study using reduction of uncertainty of underlying carbon fluxes as a metric for potential of station locations has been added:

"However, normally the metric for considering potential station location in QND studies, such as previously mentioned Nickless et al. (2020), is reduction in uncertainty of underlying carbon fluxes and tend to cluster around the station like footprints."

L423: See above. Larger sensitivity to biogenic fluxes not proven.

We are more careful in how we phrase our conclusions:

"The stations pick up signals throughout the heterogeneous European flux landscape and show a large variation of sensitivities, with a larger sensing capacity for biogenic fluxes than for anthropogenic emissions during the study period of summer year 2020. The summer is most interesting from a biosphere monitoring perspective and 2020 has also proven representative for longer time-periods in terms of our conclusions about the network"

**Technical comments**

L110 and elsewhere: 'Sect. 3.1' instead of just '3.1.

This has been implemented.

**Minor comments reviewer two.**

The abstract is overly long and not concise enough. Please condense it.

The abstract has been shortened and now reads:

"The ICOS (Integrated Carbon Observation System) network of atmospheric measurement stations produces standardized data on greenhouse gas concentrations at 46 stations in 16 different European countries (March 2023). The placement of instruments on tall towers and mountains makes for large influence regions ("concentration footprints"). The combined footprints for all the individual stations create a "lens" through which the network sees the European $CO_2$ flux landscape. In this study, we summarize this view using quantitative metrics of the fluxes seen by individual stations, and by the current and extended ICOS network. Results are presented both from a country-level and pan-European perspective, using open-source tools that we make available through the ICOS Carbon Portal. We target anthropogenic emissions from various sectors, as well as the land cover types found across Europe and their spatiotemporally varying fluxes. This recognizes different interests of different ICOS stakeholders. We specifically introduce "monitoring potential maps" to identify which regions have a relative underrepresentation of biospheric fluxes. This potential changes with the introduction of new stations, which we investigate for the planned ICOS expansion with 19 stations over the next few years.

In our study focused on the summer of 2020, we find that the ICOS network has limited sensitivity to anthropogenic fluxes, as was intended in the current design. Its representation of biospheric fluxes follows the fractional representation of land cover and is generally well-balanced considering the pan-European view. Exceptions include representation of grass & shrubland and broadleaf forests which are abundant in south-eastern European countries, particularly Croatia and Serbia. On country scale the representation shows larger imbalances, even within relatively densely monitored countries. The flexibility to consider both individual ecosystems, countries, or their integrals across Europe demonstrates the usefulness of our analyses and can readily be re-produced for any network configuration within Europe."

line 50: atmospheric needs to be mentioned here. You have mentioned land and ocean but the rest of the co2 is contributing to the atmospheric growth rate.

The reviewer is referring to the following part of the introduction which has been updated (underlined):

"Up until now, about half of the $CO_2$ humans have emitted has been taken up by land (29% of total $CO_2$ emissions 2011-2020, Friedlingstein et al., 2022) or stored in the deep ocean (26% of total $CO_2$ emissions 2011-2020, Friedlingstein et al., 2022). The other half of the anthropogenic $CO_2$ remains in the atmosphere and contributes to the atmospheric growth rate which is 2.5 ppm for 2022 according to a preliminary estimate by Friedlingstein et al. (2022)."

Line 60. Full stop missing.

A full stop has been added.

Line 93. While the exact approach used here is not awfully common, there have been a few previous studies that have examined ecosystem representation of various networks with a slightly different tool (eddy flux) and should be mentioned. Malone et al., 2022 (https://doi.org/10.5194/bg-19-2507-2022) examined the ecological representation of the NEON network of eddy flux towers over the US but for the eddy flux footprints, which are much smaller than the concentration footprints / surface influence of the ICOS network. Pallandt et al., 2022 (https://doi.org/10.5194/bg-19-559-2022) also looked at ecosystem representation in the Arctic ecosystems.

We thank the reviewer for pointing us to these interesting references which we now refer to. The following was added:

"For ecosystem sites, where fluxes rather than concentrations are measured, the "flux footprints" are small with influence mainly from the site's immediate surrounding. In this context, the idea of representation has been applied in Pallandt et al. (2022) to assess what Arctic ecosystem types are at the site locations compared to what ecosystems are found in the Arctic. Malone et al. (2022) similarly identified gaps in the U.S. NEON network based on representation of different clusters identified based on their ecological properties. In both studies, the evaluation of the network representation was subsequently used to advise on future expansion and upscaling of the fluxes to larger regions."

Line 116. Do you mean STILT v2 (Fasoli et al.,2018, https://www.geosci-modeldev.net/11/2813/2018/). If not, I'd recommend upgrading. It's much faster to produce the footprints than v1.

We thank the reviewer for the suggestion. However, STILT was set up in the ICOS Carbon Portal in collaboration with C. Gerbig at MPI for Biogeochemistry and we share input data processing. Therefore, we use the STILT version that is supported by MPI-BCG.

Line 124. I don't understand this statement. If you want to combine the footprints with biogenic $CO_2$ estimates, then you need to have at least 3 hourly (hourly would be better) time-step footprints back over the 10 days or you won't be able to interpret the $CO_2$ signals. There is a strong diurnal cycle in $CO_2$ flux from the biosphere with uptake during the day and emission at night. So you can't just take a 10 day footprint and multiply it by the net $CO_2$ from VPRM. That's not the same as combining it in a sub-diurnal way. See Commane et al, 2017, Schiferl et al., 2022 for examples. It sounds like on Line 137 that you are using the time varying footprints for the interpretation so maybe I am just misunderstanding. In that case, maybe a little re-wording/clarification would be good. Did you combine the GPP and Resp from VPRM separately? In that case, did you use a static average number for the previous 10 days? I think that would actually be ok, given what you are doing with this study. But it needs to be explained.

We thank the reviewer for this important point and useful references. We did multiply time-step aggregated footprints back over the ten days with static $CO_2$ fluxes from VPRM, but from GEE and respiration rather than the net $CO_2$. We used the flux maps matching the time the air was modelled to arrive at the receptor, which meant that we missed the strong diurnal cycle of especially GEE as pointed out by the reviewer. To do better justice to this variation, we have updated our methods:

We have re-created the study with hourly time-step footprints back over the ten days, as pointed out as preferable by the reviewer. It means updated results throughout the study, but it did not change our main conclusions (see comparison below).

Hourly backward footprints were created and saved specifically for our updated analyses and are not normally saved when the Carbon Portal Footprint Tool is run. This, in combination with the computational resources required for the updated approach, means we cannot offer this to the general users of our online tool hosted at the Carbon Portal JupyterHub (Sect. 6). Rather, we will update the tool to use a static average number for the previous ten days as suggested as an acceptable alternative by the reviewer.

The different approaches to arrive at network representation of different ecosystem fluxes are compared below for the 2022 ICOS atmospheric station network year 2020 (same information as in paper Fig. 5b, and Fig. 8b for Germany, upper bars). The fair agreement between using the static 10-day average compared to the updated approach gives confidence that it is feasible to use in the online tool.

To further understand the differences in results, we have examined the representation of coniferous forests; the higher overrepresentation in the static tool comes mainly from the night-time hours (0:00, 3:00, 6:00, 21:00) with average overrepresentations subset to the individual time-steps ranging from 12-24% compared to 13% given all hours (Table 5). There is essentially no GEE during the night-time hours when backward resolved footprints are used, whereas the static 10-day average means higher activity also during these night-hours. Footprints representing the night, when there is less mixing of the air, will generally have more local influence on the

measurements. Many of the ICOS stations, such as Hyltemossa, is close to/in coniferous forests which therefore explains the relative overrepresentation. This again stresses the advantage with the new approach suggested by the reviewer.

| | Total GEE | Broadleaf forest | Coniferous forest | Mixed forest | Other | Grass & shrub-land | Cropland | Pasture | Urban |
|---|---|---|---|---|---|---|---|---|---|
| Europe GEE "closest match" (preprint, outdated) | 3,12 | -12,56 | 8,88 | -3,72 | 56,83 | -52,93 | 8,60 | 7,21 | 59,45 |
| Europe GEE "static 10 days" (online tool) | 3,51 | -16,84 | 12,64 | -4,39 | 74,09 | -53,13 | 11,54 | -1,76 | 62,70 |
| Europe GEE "backward resolved" (paper) | 2,96 | -16,51 | 0,93 | -0,35 | 40,67 | -53,66 | 13,56 | 7,68 | 63,89 |

**Table 5. The over- (+) or under (-) representation of the current ICOS network within Europe (see paper Sect. 2.5). Results are given for GEE summer year 2020 with the different approaches to combine footprints with fluxes.**

The result presented in Table 5 will be shared in the online tool to inform users about what the alternative approach can mean for their results.

Line 178. Others.

"Other" changed to others.

The Results and discussion could be combined. I think it would read better and there is a lot of discussion in the Results section already. But that's just a suggestion.

We thank the reviewer for this suggestion but have decided to keep the sections separately. However, we have removed information in the discussion that already appears in the results to avoid redundancy.

Updated equations in accordance with the point made by reviewer one (L185ff).

We consider the model domain 15°W to 35°E and 33°N-73°N:

For a given country or region, $C$, we can look at $C_{i,j}$ which is the fraction of the country/region in a given grid cell $(i, j)$. We consider specific land cover types, $LC$, and use $LC_{i,j}$ which is the fraction of land cover within a given grid cell.

We establish the network view ($N_{i,j}(T)$) and equal view ($NEQ_{i,j}(T)$) of the flux land scape ($GEE_{i,j}(t_k)$): For each grid cell $i, j$ and hour $t_k$ leading up to when the air arrives at the receptor (T) where hour $t_k = t_1, t_2, t_3, \ldots, t_{240}$ (here $t_{240}$ is T, the time the air arrives at the receptors of the stations in the network).

$$N_{i,j}(T) = C_{i,j} \cdot LC_{i,j} \sum_{k=1}^{240} NFP_{i,j}(t_k) \cdot \left| GEE_{i,j}(t_k) \right| \tag{1}$$

$$NEQ_{i,j}(T) = C_{i,j} \cdot LC_{i,j} \sum_{k=1}^{240} \frac{\sum_{m,n} C_{m,n} \cdot NFP_{m,n}(t_k)}{\sum_{m,n} C_{m,n}} \cdot \left| GEE_{i,j}(t_k) \right| \tag{2}$$

Where *C* and *LC* are the fractional country grid of selected country or region. *NFP* is the network footprint (see Sect. 2.4) and *GEE* is the flux map and these change with time ($t_k$). *m,n* are sum indices for the that run over all cell coordinates in the model grid.

The relative flux representation (*REP(T)*), used in Fig. 5b and Fig. 8b is the ratio between the total sensing within the grid cells of the network view ($N_{i,j}(T)$) and the equal view $NEQ_{i,j}(T)$).

$$REP(T) = \frac{\sum_{i,j} N_{i,j}(T)}{\sum_{i,j} NEQ_{i,j}(T)} \tag{3}$$

The relative monitoring potential maps (*MP(T)*, used in Sect. 3.2 and Sect. 3.3) show the difference in the sensing between the network view and the equal view within the individual grid cells of the model.

$$MP(T) = max\left( NEQ_{i,j}(T) - N_{i,j}(T), 0 \right) \tag{4}$$

Cells where the equal view ($NEQ_{i,j}$) is greater (more uptake), than the network view ($N_{i,j}$) will have positive values in the relative monitoring potential map (*MP*), and only these are displayed. Monitoring potential becomes especially high in areas where the current network is relatively blind, and the activity of the specific flux is relatively high. For the monitoring potential maps for the extended network, the equal view ($NEQ_{i,j}$) is kept the same as for the current network to facilitate effective comparison between the maps.